# Dynamical stability and mechanical impedance are optimized when manipulating uncertain dynamically complex objects

Rakshith Lokesh[1,2]*, Dagmar Sternad[1,2,3,4]

**1** Department of Biology, Northeastern University, Boston, Massachusetts, United States of America,
**2** Department of Electrical and Computer Engineering, Northeastern University, Boston, Massachusetts,
United States of America, **3** Department of Physics, Northeastern University, Boston,
United States of America, **4** Institute for Experiential Robotics, Northeastern University, Boston,
Massachusetts, United States of America

* r.lokesh@northeastern.edu

Dynamical stability and mechanical
impedance are optimized when
manipulating uncertain dynamically complex
objects. PLoS Comput Biol
21(12): e1013797. https://doi.org/10.1371/
journal.pcbi.1013797

University, BELGIUM

**Peer Review History:** PLOS recognizes
the benefits of transparency in the peer
review process; therefore, we enable the
publication of all of the content of peer
review and author responses alongside
final, published articles. The editorial
history of this article is available here:
https://doi.org/10.1371/journal.pcbi.
1013797

## Abstract

Manipulating complex objects is ubiquitous in our daily activities, such as donning
a jacket or carrying a cup of coffee. However, such non-rigid objects easily become
unstable: When carrying a cup of coffee, the coffee could slosh unpredictably and
spill out of the cup. It remains unclear what motor control strategies ensure stabil-
ity, especially when the physical properties of the object, like the amount of liquid
in the cup, are unknown. The task of transporting a 'cup of coffee' was simplified
to transporting a virtual cup with a sliding ball inside, modeled as a cart-pendulum
system. Participants were instructed to 'jiggle' the cup in one dimension to prepare
the cup and ball states for the ensuing continuous rhythmic movement. To introduce
uncertainty regarding the object's properties, the pendulum's length was manipulated
either to 1) change randomly from trial to trial, or to 2) remain constant across trials.
We measured the ball's angle at the end of preparation and the cup's oscillation fre-
quency during the rhythmic portion of the trial. Grip force on the robot handle served
as proxy for mechanical impedance of the arm. The results supported three predic-
tions: 1) When dynamic uncertainty was high, object preparation was important to
stabilize the transient dynamics; stability increased during preparation and humans
prepared longer when the dynamics was uncertain. 2) Humans maximized dynamic
stability by flexibly covarying system initialization and cup frequency; dynamic sta-
bility matched participant behavior better than magnitude or smoothness of force. 3)
Humans increased their arm impedance to accommodate uncertain dynamics, while
the net force applied on the cup remained the same. Feedforward simulations using
an impedance controller and stochastic open-loop optimal control corroborated these
findings, further revealing that participants' selection of preparation and interaction
frequencies also minimized mechanical impedance. In sum, humans used prepara-
tion and interaction strategies to optimize the mechanical impedance and dynamic

**Data availability statement:** All relevant data are within the manuscript and at the following public repository https://osf.io/fuz8w/?view_only=53d9b6c87e7f4337b50d7ae635683600.

**Funding:** DS received funding from the National Institute of Health grant NIH-R01-CRCNS-NS120579 and grant NIH-R37-HD087089. The funders had no role in study design, data collection and analysis, decision to publish, or preparation of the manuscript. Both authors received salary from the grants funded by the above organizations.

stability of the hand-object interactions. These results may inform approaches in robotic control and rehabilitation.

## Author summary

Humans use and interact with a wide variety of objects for their daily lives. Many activities involve non-rigid and deformable objects, like using a towel, tying shoelaces, or sipping from a cup of coffee. Although these actions appear easy and straightforward for healthy individuals, the challenges become evident when neurologically impaired people or robots handle such objects. Leading a cup of coffee to one's mouth can cause unpredictable sloshing and spilling of the coffee. How do humans manipulate such objects while ensuring stable behavior? Motivated by the task of moving a cup of coffee, participants rhythmically translated a virtual cup with a ball rolling inside it. The physical characteristics of the object were randomly manipulated without informing the participants. They were allowed to prepare the cup and ball prior to rhythmic manipulation. Results showed that participants took more time to prepare the object when the cup-ball behavior was not known in advance. Interestingly, they flexibly prepared the state of the ball in relation to the frequency of the cup movements. Together with higher grip force, particularly when facing unknown dynamics, this ensured stability of the cup and ball. These findings were supported by computer simulations that employed a human-inspired controller. These insights into human object handling may inform the development of control algorithms for robotic manipulation and inform rehabilitation strategies for patients with neurological injuries.

## Introduction

Humans use a towel, tie their shoelaces, or carry a cup filled with coffee with relative ease. Yet, manipulating such non-rigid objects presents complex dynamical challenges. For instance, leading a cup of coffee to one's mouth can be difficult due to the lack of direct control over the coffee's motion, creating unpredictable dynamics that may lead to spilling of the coffee [1,2]. The intricacies of this interactive dynamics become evident in people with neurological impairments and also pose problems for the manipulation of objects in robots. Adding to these difficulties is that in daily life the physical properties of such objects often change, like the shape of the cup or the amount of coffee left in the cup. To address these challenges, humans can rely on motor control strategies that extend beyond those used to manipulate rigid objects with fully actuated forces.

Computational motor control research has primarily focused on unconstrained arm movements or the manipulation of rigid objects, which are essentially extensions of the hand mass. Most of the studies on such simple object manipulation have focused on cognitive aspects, like perception and representation of object features [3,4], size-weight illusion [5,6], planning of finger configurations for different object geometries [7–9] or grasping forces necessary to transport different object shapes [10,11]. To date, only a few studies have explored human dynamic interactions with

more complex systems, such as a mass-spring [12–14], a double-pendulum [15], a whip [16,17], or a cart-pendulum system [18,19], that possess internal degrees of freedom that are not directly controlled. These studies have shown that factors, like the stability of the dynamics are of concern, due to the unpredictable feedback forces and the potential for chaotic behavior [20,21].

The dynamical stability of nonlinear systems is influenced by characteristics of the forcing signal, such as its amplitude and frequency [22]. For example, small changes in forcing frequency near the resonance frequency can cause rapid shifts in relative phase in the system's output relative to its input. Using a virtual cart-pendulum task inspired by carrying a cup of coffee, visualized as a cup with a rolling ball inside, participants were shown to avoid driving the cup close to the anti-resonant frequency (resonance frequency of the pendulum) to prevent unpredictable forces and unstable ball movement [23,24], thereby reducing the need for continuous sensory feedback-based control [18]. By establishing stable dynamics, humans avoid relying on feedback-based control, which can be challenged by neural transmission delays, uncertain or simplified internal models, and unmodeled environmental forces [12,18,25,26]. Collectively, these studies suggest that, when interacting with objects that exhibit complex dynamics, humans select interaction strategies to increase the stability of the dynamics.

Stability of a nonlinear system is also sensitive to its initial conditions, often dubbed the 'butterfly effect'. Indeed, humans frequently prepare and adjust initial conditions of complex dynamical objects to simplify the subsequent control. In several daily-life and skill-based activities, we instinctively prepare the state of an object before manipulating it in a desired way, such as casting cloth in the air before spreading it on a table. Using a cart-pendulum system, i.e., the cup-ball task that is also used in this study, our group's previous research showed that humans adjust the angle of the ball before executing rhythmic back-and-forth movements of the cup, ensuring simple and stable ball dynamics [27]. Building on this finding, follow-up work demonstrated that humans flexibly and nonlinearly modulated both the ball's initial conditions and the cup's frequency to achieve stable dynamics of an object with uncertainties [28]. However, it remains unclear how humans prepare the object and select interaction frequencies when the object's dynamics is uncertain.

One potential strategy for coping with uncertainty in interaction dynamics is through the regulation of mechanical impedance. For example, the antagonistic muscle pairs around the shoulder, elbow and wrist joints can be co-contracted to increase the endpoint stiffness at the hand [29]. This increase in stiffness helps stabilize movements in uncertain or perturbing environments [30–32], dampen motor variability caused by intrinsic noise [33,34], and buffer unexpected environmental disturbances [30,35]. The central nervous system (CNS) has also been shown to plan the impedance in a feedback-based manner, eliminating the need for sensory-based corrections during deviations from desired movements [25,36,37]. More recent computational studies have used stochastic optimal open-loop control to model impedance planning via muscular co-contraction, replicating findings from classic motor control experiments [38,39]. In human interactions with complex objects, simple feedforward controllers [24,26] and optimal control models [40] that incorporate impedance have been used to replicate kinematic features of movement. However, these models did not allow impedance to be modulated in real time within a single interaction bout, nor did they account for uncertainty in object dynamics. Therefore, it is crucial to explore how mechanical impedance addresses dynamical uncertainties during interactions and how it relates to object preparation and the choice of interaction frequency.

The present study used the virtual cup-ball paradigm to assess how humans deal with uncertain dynamics. In the experiment, the dynamics of sloshing coffee was simplified by modeling the coffee as a ball (a point mass) sliding inside a semi-circular cup. This rendering corresponded to the dynamics of a cart-pendulum system. Building on our previous work [28], we investigated how participants prepared the object states, selected interaction frequencies and modulated impedance, while manipulating the cup-ball system with uncertain dynamics. Participants were encouraged to prepare and set "initial conditions" of the system before executing rhythmic cup movements along a horizontal line. We tested two experimental conditions: (i) random—where parameters of the cup-ball system changed randomly between trials without giving explicit cues to the participants, and (ii) blocked—where the parameters remained constant across a set of trials.

We tested three main predictions: 1) Object preparation is important to stabilize the object and task performance when dynamic uncertainty is high, because more explorations would be required to find and establish dynamic stability when the system parameters are uncertain. Therefore, preparation will take longer in the random condition. 2) Humans maximize stability of dynamics by flexibly covarying the system's initial conditions and the following cup frequency, because initial conditions and interaction frequency affect stability of the system. Therefore, relative phase variability (stability) explains participant choice of preparation and interaction frequency better than applied force or smoothness of applied force. 3) Humans increase their arm impedance, but not the net force, to account for uncertain dynamics; arm impedance likely helps attenuate the variability in the system states. To buttress our empirical results, we employed stochastic open-loop optimal control to predict impedance features for both the random and blocked conditions. In addition, we characterized the dynamic stability and mechanical impedance of the system in relation to the preparation frequency and the interaction frequency.

## Methods

### Ethics statement

Participants gave written informed consent as approved by Northeastern University's Institutional Review Board (IRB#:10-06-19).

### Participants

12 individuals (19 to 35 years, Mean: 25.8, Std. Dev.: 2.8, 7 females) participated in the experiment. All were right-hand dominant and had no history of neurological or biomechanical issues. Experimental procedures were explained to each participant prior to the study.

### Experimental task and its dynamical model

Similar to prior experiments [18,24,27], participants were tasked with transporting a "cup of coffee", simplified to a ball sliding inside a circular arc (Fig 1A). This cup was constrained to move on a horizontal line delimited by two target boxes. The cup-ball system was mechanically equivalent to a cart with a suspended pendulum where the ball corresponded to the pendulum bob and the cup was the semi-circular path of the pendulum bob. The angular position of the ball on the semi-circular arc was equivalent to the angle of the pendulum. Despite this simplified representation, the essential features of underactuation and nonlinearity present in the real dynamics of "carrying a cup of coffee", were retained. The following dynamical equations describe the cup-ball system:

$$(m_c + m_b)\ddot{x} = m_b l(\dot{\theta}^2 sin\theta - \ddot{\theta} cos\theta) + F_{applied}$$
$$= F_{ball} + F_{applied} \tag{1}$$

$$\ddot{\theta} = -\frac{\ddot{x}}{l}cos\theta - \frac{g}{l}sin\theta \tag{2}$$

where $x$ is the cup position and $\theta$ is the ball angle. The ball position at the bottom of the cup (downward vertical orientation of the pendulum bob) was defined as 0 deg, with counterclockwise direction defined as positive. Similar to [27], the mass of the cup was $m_c = 2.4\ kg$, the mass of the ball was $m_b = 0.6\ kg$, and the gravitational acceleration was $g = 9.81 m/s^2$. $F_{applied}$ is the force applied by the participant's hand on the cup. $F_{ball}$ is the force applied by the ball on the cup. Different from our prior experiment [27], the length of the pendulum $l$ was varied in the experiment between three values: 0.3 m, 0.6 m, or 1.2 m.

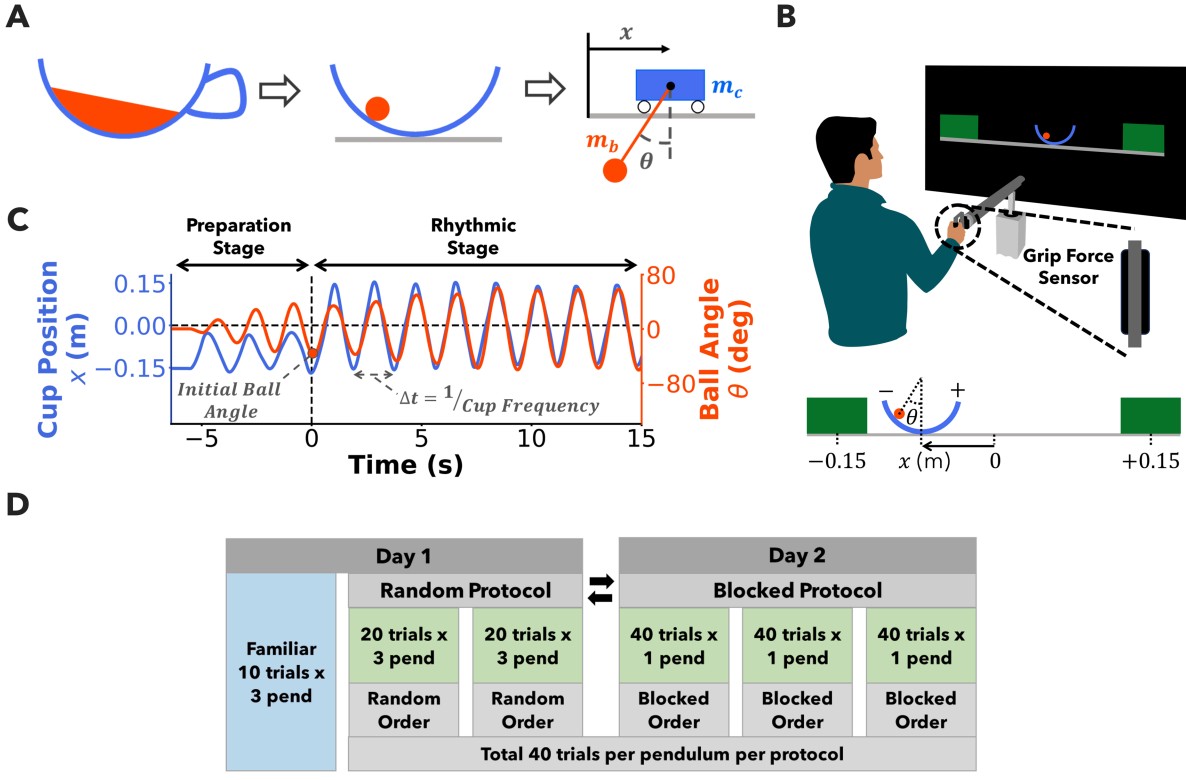

**Fig 1. Experimental task and protocols. A)** The task of carrying a 'cup of coffee' was simplified to a ball sliding inside a semicircular cup and modeled as a cart-pendulum system. **B)** The cup-ball task was displayed on a screen. Participants controlled the cup shown on a screen by moving the handle of a robotic manipulandum. The cup was constrained to move horizontally in 1-D delimited by two target boxes. **C)** Exemplary trial. Participants 'jiggled' the cup to explore and prepare the cup-ball system in the 'Preparation Stage' before executing rhythmic oscillations of the cup between the two boxes at their choice of frequency in the 'Rhythmic Stage'. The initial ball angle represented preparation of the cup-ball system and the cup frequency in the rhythmic stage represented continuous interaction. **D)** Three different pendulum lengths were used to manipulate the uncertainty of the ball dynamics: short (0.3 m), medium (0.6 m), and long (1.2 m). Two experimental protocols were tested: i) Random protocol: the pendulum length was varied from trial to trial within a block without providing any explicit cues, and ii) Blocked protocol: the pendulum length remained constant within a block of trials. Each participant completed 40 trials per pendulum in both random and blocked protocols. The order of random and blocked protocol was counterbalanced as indicated by the bidirectional arrows. The grip force exerted by the participant on the handle was measured using a grip force sensor as shown in **B**. The grip force was used as a proxy for mechanical impedance of the arm.

## Experimental setup and procedures

The task was realized in a virtual environment and displayed on a large projection screen facing the participant. The cup was displayed as a blue circular arc and the ball as a red circle (Fig 1B and 1C). A thin grey horizontal line representing the floor was bounded by two rectangular green target rectangles, Box A and Box B, their centers at $-0.15\,m$ and $+0.15\,m$, with the midpoint defined as zero. Importantly, regardless of the pendulum length, the visual scale of the cup and ball remained the same (cup radius: 22.5 cm, ball radius: 2.2 cm) to conceal the specific system to the participant. Participants sat on a chair in front of the screen and gripped the handle of an admittance-controlled robotic manipulandum (HapticMaster, Motekforce, Netherlands, Fig 1B). Participants controlled the position of the cup $x$ by moving the robotic handle laterally. The position of the chair was adjusted such that the participant could comfortably grasp the handle and move the cup between the two rectangular boxes. The motion of the robotic handle was limited to horizontal translations. The ball force $F_{ball}$ was fed back to the participant's hand via the robotic handle. The system equations, system parameters, task parameters, and the task instructions were the same as in a prior experiment [27]. The applied force on the cup

$F_{applied}$, the position $x$, velocity $\dot{x}$ and acceleration $\ddot{x}$ of the cup, and the computed angular position $\theta$, velocity $\dot{\theta}$, and acceleration $\ddot{\theta}$ of the ball were recorded at 120 $Hz$, using a custom-written C++ program. The grip force exerted by the participants on the handle was measured using a grip force sensor [41]. The net force applied by the participant on the handle was measured using a six-axis load sensor.

At the beginning of each trial, the cup was on the left in Box A with the ball at rest at $\theta$ = 0 deg. Each trial had two stages: preparation and rhythmic stage. In the preparation stage, similar to prior experiment [27], participants were encouraged to explore the cup and ball dynamics by 'jiggling' the cup anywhere on the left side of the right green box, Box B (0.15 $m$). The exploration time was not restricted. When participants were ready to start the rhythmic stage, they moved the cup into Box B, and then continuing with rhythmic back and forth movements between the two target boxes. A bell sound was played when participants reached Box B for the first time. For the rhythmic stage, participants were instructed to oscillate the cup with a consistent frequency of their choice without losing the ball for the next 15 seconds. After 15 s a bell sound indicated the end of the trial. Fig 1C shows an example trial of cup position and ball angles across the entire trial including the preparation and rhythmic stages.

## Experimental protocol

Two experimental protocols (Fig 1D) were tested: random protocol - the three pendulum lengths were varied randomly between trials (3 blocks of 40 trials, 120 trials in total), and blocked protocol - the pendulum length remained constant within a block of trials (with 40 trials x 3 pendulum, 120 trials in total). Participants completed the experiment over two experimental sessions on consecutive days, each lasting approximately one hour (Fig 1D). At the beginning of the first session, participants completed 10 familiarization trials with each pendulum ($l$=0.3, 0.6, or 1.2 m) in a blocked fashion (counterbalanced between participants). After these familiarization trials, each participant completed the random protocol and the blocked protocol over two days, with one protocol per day. The order of the two protocols was counterbalanced between participants. Such a within-participant design allowed a more direct comparisons of the two protocols' effect on arm impedance, applied force, and cup-ball kinematics. In the random protocol, no external cues were provided that revealed the pendulum length in a given trial. For the blocked protocol, participants performed 3 blocks of 40 trials, with pendulum conditions presented in a blocked order. The pendulum condition order was counterbalanced between participants. Familiarization trials were excluded from the data analysis.

## Data analysis

**Parsing into preparation and rhythmic stage.** For each trial, the time at which the cup velocity was first 0 deg/s before the start of the rhythmic stage, i.e., before moving the cup into Box B, was identified. This time point, defined as 0 time, represented the end of the preparation stage and start of the rhythmic stage. This time instant is different from the start of each experimental trial that determined the start of the preparation stage.

**Dynamic stability - Relative phase variability.** Relative phase between the cup position and ball angle characterized how participants controlled the cup and ball relative to one another within each trial. To obtain relative phase, the instantaneous unwrapped (Hilbert) phases of cup position and ball angle were subtracted from each other. A relative phase of 0 or 180 deg indicated in-phase or anti-phase relation between the cup and ball, respectively. The cup phase, ball phase, and their relative phase during the rhythmic stage from one trial are shown in Fig 2. Variability of the relative phase in the rhythmic phase was used as a proxy to quantify the degree of stability of the relation between the cup and ball kinematics. This metric was previously used in work on rhythmic coordination modeled as coupled oscillations [42]. Since the relative phase is a circular variable, the circular variance of the relative phase was computed and denoted as relative phase variability [43]. Relative phase variability ranged between 0 and 1, and smaller values (less spread) indicated greater stability. The circular variance for the example trial in Fig 2 is 0.002.

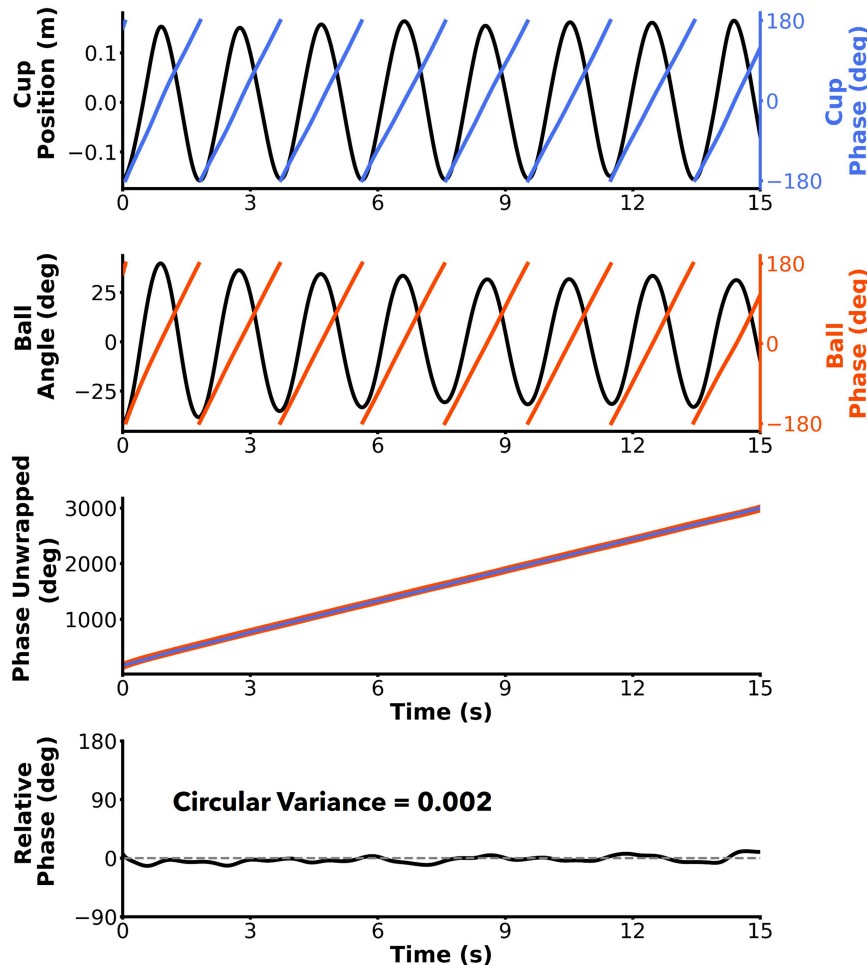

**Fig 2**. **Relative phase variability.** Exemplary trial visualizing the computation of relative phase between the cup and ball. The phases of the cup and ball were computed using the Hilbert transform. The phases were then unwrapped and subtracted to obtain the relative phase or phase difference. The cup and ball oscillate in phase, as indicated by the relative phase hovering around 0 deg, a relative phase of 180 deg represents anti-phase oscillations. The circular variance of relative phase that represents relative phase variability is 0.002.

**Preparation duration and time to reach stability.** The time from the start of the trial to the start of the rhythmic stage was labeled preparation duration. The time taken to stabilize relative phase was calculated using a windowed procedure on relative phase variability. We implemented a time window with a movable left bound and the right bound fixed to the end of preparation stage. Relative phase variability within the time window was computed each time the left bound was moved backward by one time sample. The time point at which the relative phase variability first increased above a threshold of 0.1 was identified. The time from the beginning of the trial to this point was considered as the time to stable relative phase. The threshold was selected based on the relative phase variability heatmaps from simulations in Fig 6C. A threshold of 0.1 indicated stable (darker) regions for all pendulums.

**Initialization of the cup-ball system - Initial ball angle.** The ball angle at the start of the rhythmic stage, time 0, was defined as the initial ball angle. The initial ball angle represented the outcome of the preparation stage for the subsequent rhythmic stage [27].

**Rhythmic interaction - Cup frequency.** A primary variable of interest was the continuous frequency of the cup to allow quantifying deviations from a constant frequency. The frequency at which the participants chose to oscillate the cup

during the rhythmic phase represented the driving frequency. Using the Hilbert Transform, the instantaneous phase of the rhythmic cup position was computed which was then unwrapped and numerically differentiated to render instantaneous frequency [44].

**Impedance estimated from isometric grip force.** Previous work has provided evidence that the mechanical impedance of the arm is important during interactions with the cup-ball system [24,40] and can potentially be modulated to improve the stability of interactions with the cup. Prior work has shown that the grasp force was used to regulate arm impedance to improve interaction stability [45] and manipulation accuracy [46] when handling objects in the environment. It has also been experimentally verified that grasp force was significantly correlated to the end-point mechanical impedance characteristics of the arm [47–49]. Hence, the force exerted by the participants to grip the handle was used as a placeholder for the mechanical impedance of the arm. The graspable sensor at the handle measured the sum of net applied force and the gripping force [41]. Hence, the applied force on the handle was subtracted from the graspable sensor reading to obtain the gripping force.

## Simulations of a simple control model

To gain insight into the control of the cup-ball system, we used a first-order impedance controller to model a participant moving the cup [24,27]. The physiological premise behind such a controller is that muscles act like nonlinear springs with modifiable rest length enabling changes to the equilibrium position of the joint [50]. As a result, equilibrium trajectories or desired trajectories can be set at the end point of the arm, i.e., the hand.

$$F_{applied} = K(x - x^d) + B(\dot{x} - \dot{x}^d) \tag{3}$$

The desired cup position was set to be $x^d = -ACos(2\pi ft)$ to adequately capture the instruction to move the cup rhythmically between the target boxes, where $A = 0.3$ m, $f$ was the oscillation frequency of the cup, and $t$ was the simulation time. The simulation was repeated sweeping over different combinations of cup frequencies and initial ball angles. The initial ball angle was varied between $\theta_0 \in \{-90\ deg, 90\ deg\}$ and cup oscillation frequency was varied $f \in \{0.3\ Hz, 1\ Hz\}$. The initial ball angular velocity was set to $-4deg/s$ corresponding to the mean initial ball velocity at the start of the rhythmic stage across all participants. The total simulation time was $15\ s$ as in the experiment, and the step size was $\Delta t = 10\ ms$. A 4th-order Runge-Kutta integrator was used to forward simulate the system upon applying the control input $F$ starting from the initial conditions. For each simulation run, the relative phase and relative phase variability was computed as described earlier. We then plotted a heatmap of relative phase variability as a function of cup frequency and initial ball angle.

We then tested whether the cup frequencies and ball angles covaried to satisfy not only relative phase variability, but also other objectives, such as expended force, smoothness or risk of losing the ball. Hence, we also computed mean absolute force, log dimensionless jerk of applied force, and risk of ball escape. The mean absolute force was computed by obtaining the mean of the force applied by the participant on the handle. The log dimensionless jerk of the applied force was calculated as follows [51]:

$$Jerk\ Applied\ Force = -log\left(\left|\frac{(t_{end} - t_0)^3}{F_{peak}^2} \int_{t_0}^{t_{end}} \ddot{F}^2 dt\right|\right) \tag{4}$$

where, $F$ is the applied force, $F_{peak}$ is the peak force, $(t_0 = 0,\ t_{end} = 15)$ are the time start and end of the force signal.

The risk of the ball escaping from the cup was computed using energy measures [27]. The escape energy was the maximum possible total energy of the ball before it escaped beyond the rim of the cup ($\theta_{rim} = 60\ deg$). The energy margin

between the total energy of the ball and the escape energy at any given time point in the trial was calculated as follows.

$$E_{total} = E_{kinetic} + E_{potential} = \frac{\dot{\theta}^2 l^2 m}{2} + mgl(1 - \cos\theta)$$
$$E_{escape} = mgl(1 - \cos\theta_{rim})$$
$$E_{margin} = E_{escape} - E_{total}$$

(5)

The risk of ball escape for the full trial duration $T$ was calculated as follows:

$$Risk = 1 - \frac{1}{TE_{escape}} \int_0^T E_{margin} dt$$

(6)

Mean absolute force served as a measure of effort, log dimensionless jerk as a measure of smoothness of applied force, and risk of the ball exceeding the cup's rim, i.e., loss of ball and failure of a trial.

For each variable, we used the Kullback-Leibler divergence to test how well the simulated heatmaps explained participants' choices of cup frequency and initial ball angle. We computed KL-divergence between the probability density of the heatmap and the 2D histogram of participant data for each pendulum and protocol.

**Impedance predictions using optimal control simulations**

The fundamental assumption of our model was that the central nervous system sets and modulates impedance to account for uncertainty in the interaction dynamics. The model focused on impedance control via descending motor commands based on the knowledge of uncertainty in the dynamics of the task. We assumed that participants recognized the uncertainty in the task, preplanned impedance, and tuned the impedance in a feedforward fashion. Thus, uncertainty was added to the pendulum length to model trial-to-trial variations in pendulum length for the random protocol. Participants control of the cup was modeled by a controller consisting of impedance and feedforward force terms. The impedance gains and the feedforward forces were optimized using the Stochastic Open Loop Optimal Control framework while enforcing constraints and costs according to the experimental task.

**Modeling the cup-ball system with uncertainty in pendulum length.** The uncertainty due to the random variation of pendulum length from one trial to the next was modeled by adding noise to the pendulum length. Uncertainty arising from trial-by-trial variations in task parameters has previously been modeled as additive Gaussian noise in force field adaptation tasks [38,39]. Thus, pendulum length $l$ in the cup-ball model was converted to a state variable buffeted by Gaussian noise $\varepsilon$. For the random protocol, the pendulum length was considered as a discrete random variable with values 0.3, 0.6 and 1.2, each occurring with a probability of 1/3. Therefore, the mean pendulum length $l_{mean} = 0.7\ m$ and the standard deviation of the pendulum length was $\sigma_l = 0.37\ m$. To simulate the blocked protocol, the mean pendulum length $l_{mean}$ was either 0.3 $m$ or 0.6 $m$, or 1.2 $m$ with a smaller standard deviation of $\sigma_l = 0.1$. The Ornstein-Uhlenbeck process with a reversion rate of $\alpha = 100$ and standard deviation $\sigma_{ou}$ was used to model the dynamics of $\dot{l}$. The Ornstein-Uhlenbeck process led to a variance in $l$ equal to $\sigma_l^2 = \frac{\sigma_{ou}^2}{2\alpha}$. $d\varepsilon$ represents the increment of a standard Brownian process. The full state space dynamics was as follows:

$$\dot{x} = v$$
$$\dot{v} = \frac{m_b l(\omega^2 \sin\theta + \frac{g}{l} \sin\theta \cos\theta) + F_{applied}}{m_c + m_b \sin^2\theta}$$
$$\dot{\theta} = \omega$$

(7)

$$\dot{\omega} = -\frac{\dot{v}\cos\theta}{l} - \frac{g\sin\theta}{l}$$
$$\dot{l} = -\alpha(l - l_{mean}) + \sigma d\varepsilon$$

**Defining the optimal control problem.** A first-order impedance controller similar to Eq 3 [24] was used to model the hand applying forces to the cup $F_{applied}$:

$$F_{applied} = K(x - x^d) + k_b K(v - v^d) + F_{ff} \tag{8}$$

where $K$ represents the end-point stiffness, $k_b K$ represents the end-point viscosity of the arm, and $x^d$, $v^d$ are the desired states for the cup/hand. The impedance term $K$ captures the mechanism of coactivating muscles of the arm to increase end-point mechanical impedance preventing deviation from the desired trajectory. The damping coefficient was set to be proportional to the stiffness parameter, with a proportionality constant $k_b = 0.2$ to simplify numerical optimization. The feedforward force $F_{ff}$ captured reciprocal muscle activation in the arm used to achieve or alter the desired trajectory of the cup [29,52,53].

The controller parameters $K$ and $F_{ff}$ could vary over time within a trial to achieve the task requirements. To ensure smooth changes in force and impedance parameters, the second derivatives of $K$ and $F_{ff}$, i.e., $\ddot{K}$ and $\ddot{F}_{ff}$ were used as pseudo-controls. Thus, $K$ and $F_{ff}$ and their first derivatives $\dot{K}$ and $\dot{F}_{ff}$ were augmented as state variables. The full state space model could be compactly written as follows:

$$dX_t = f(X_t, u_t, t)dt + G_t d\varepsilon_t$$
$$X_t = [x_t, v_t, \theta_t, \omega_t, l_t, F_t, K_t, \dot{F}_t, \dot{K}_t] \tag{9}$$
$$u_t = [\ddot{K}_t, \ddot{F}_{ff_t}]$$

where $X \in \mathbb{R}^9$ is the state vector, $u \in \mathbb{R}^2$ is the control vector, matrix $G \in \mathbb{R}^{9 \times 1}$ accounts for the additive noise on pendulum length, and $d\varepsilon_t$ is scalar Gaussian noise.

The problem was to find the deterministic controls $\ddot{K}$ and $\ddot{F}_{ff}$ to control the state of the cup and ball according to the task requirements. As described above, the primary task goal was to move the cup rhythmically at the desired frequency $f$ and amplitude $A = 0.15\,m$ during the rhythmic stage. Similar to the forward simulations, the desired position and velocity trajectory of the cup could be designed as $x^d = A\cos(2\pi ft)$ and $v^d = -2\pi fA\sin(2\pi ft)$, respectively. The duration of the rhythmic stage was set to 6 s.

Prior to the rhythmic stage, it was necessary to set the state of the ball to some desired $\theta$ and $\omega = 0\,rad/s$, and the state of the cup to $x = -0.15\,m$ and $v = 0\,m/s$. The ball and cup states during the preparation stage could vary freely, except at the end. The duration of the preparation stage was set to 2 s. There were no other requirements on the state of the cup or the ball. Additionally, the uncertainty in pendulum length could lead to variability in the states of the cup and the ball. Hence, it was necessary to minimize the variance of the cup position and velocity whenever constraints were imposed on the states. Considering the desired state trajectory and minimization of variance around the desired states, a cost function was assembled as follows:

$$J = (X_f - X_f^d)^T Q_f (X_f - X_f^d) + \mathrm{Tr}(Q_f^v P_f) +$$
$$\int_0^t ((X_t - X_t^d)^T Q_t (X_t - X_t^d) + \mathrm{Tr}(Q_t^v P_t) + u_t^T R_t u_t)dt \tag{10}$$

where the subscripts $t$ and $f$ represent trial time and the end of trial (terminal) time, respectively. $X$ and $X^d$ are the state vector and desired state vectors, respectively, $P$ is the state covariance matrix, $Q$ is the state cost weighting matrix, $Q^v$ is the covariance cost weighting matrix. $u$ is the control vector, $R$ is the control cost weighting matrix, and Tr is the trace of the matrix. The weighting matrices $Q_f$, $Q_f^v$, $Q_t$, $Q_t^v$ and $R_t$ were designed according to the goals of the task. The cost function and matrices are described in detail in the supplementary document S1 Text.

**Deterministic optimal control solution.** We used the Stochastic Optimal Open Loop control (SOOC) framework [54] to obtain the deterministic time-varying control vector $u$, subject to the cost function $J$. An approximate solution to the stochastic optimal control problem was obtained by solving the associated deterministic optimal control problem. The deterministic optimal control problem was defined by the propagation of the mean and covariance of the states. The dynamics of the mean and variance of the states for a state space model of the form Eq 9 are given by:

$$\dot{m}_t = f(m_t, u_t, t)$$
$$\dot{P}_t = F(m_t, u_t, t)P_t + P_t F(m_t, u_t, t)^T + G_t G_t^T$$

(11)

where $m_t \in \mathbb{R}^9$ is the time-varying mean state vector, $P_t \in \mathbb{R}^{9 \times 9}$ is the time-varying state covariance matrix, and $F(m_t, u_t, t) = \frac{\partial f(m_t, u_t, t)}{\partial t}$.

The solution to the above defined deterministic optimal control problem was obtained numerically using Sequential Least Squares Programming (SLSQP) in *Python*. The initial guess for the pseudo-control $\ddot{K}$ was zero for all $t$. The initial guess for $K$ and $\dot{K}$ at $t = 0$ were 10 and 0, respectively, to prevent divergence to negative values. An initial guess for $\ddot{F}_{ff}$ was obtained using the impedance controller defined in Eq 3 using a small gain $K = 10$ to improve the convergence speed of the numerical optimization. We repeated the optimization procedure for various combinations of the desired cup frequency $f \in \{0.3, 0.8\}$ in the rhythmic stage and the initial ball angle $\theta \in \{-70, 70\}$ at the end of the preparation stage.

## Statistical analysis

The experiment adopted a within-subject design such that each participant completed all experimental conditions. Prior to specific tests, normality of the dependent variables was checked using the Shapiro-Wilk test. In case normality was violated, the logarithm of the respective variable was calculated. To test Prediction 1, we analyzed the effect of preparation duration, and time to stable relative phase, using a 2 x 3 ANOVA with experimental protocol (blocked, random), and pendulum length (short, medium, and long). The cup frequency distributions were compared across the three pendulum conditions using Kolmogorov-Smirnov tests. To test Prediction 2, the distributions of initial ball angle and cup frequency were visually compared to the heatmaps of the forward simulations of relative phase variability. Kullback-Leibler (KL) divergence was computed between experimental data and the forward simulation heatmaps of relative phase variability. The alternative objectives, absolute force, force smoothness and risk of ball escape, were analyzed in the same way. To test Prediction 3, a two-way ANOVA evaluated the effect of protocol and pendulum length on grip force (as proxy for impedance). We also reported the effect sizes using the partial Eta-squared ($\eta^2$) measure for reported statistical effects. Post-hoc tests were Bonferroni corrected. The *aov* function in *R* was used to run the ANOVAs. We used a significance threshold of $\alpha = 0.05$.

## Results

### Relative phase variability across protocols and pendulum conditions

To start, relative phase was plotted over trials for the different pendulum and protocol conditions (Fig 3A). For the short pendulum condition, participants maintained a relative phase value close to 0 deg, which corresponds to in-phase oscillations between the ball and the cup. For the medium and long conditions, the relative phases also showed anti-phase relations with values around 180 deg. Circular variance of relative phase (relative phase variability) was used to quantify

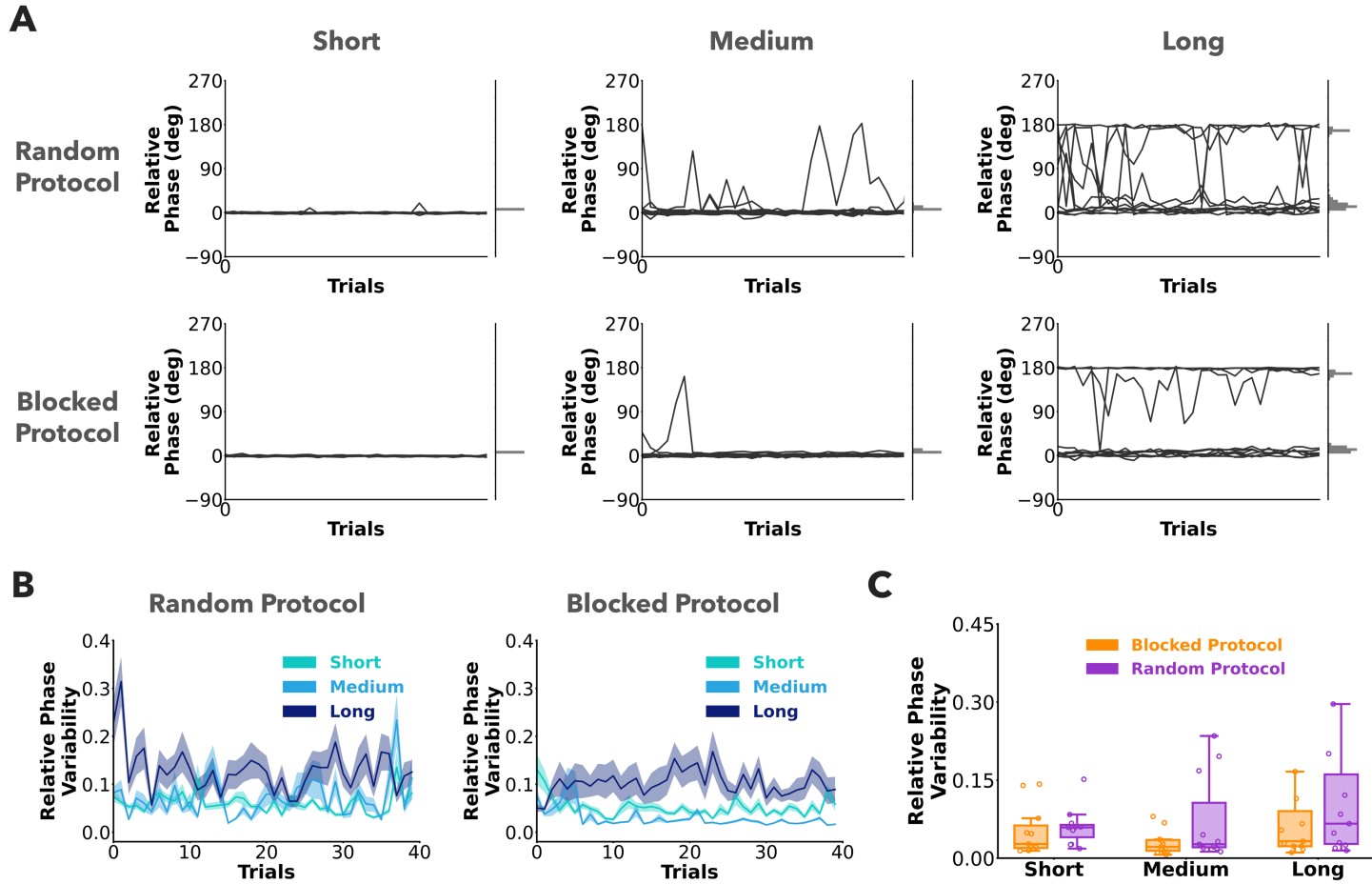

**Fig 3**. **Relative phase. A)** Relative phase plotted against trials for the three pendulum conditions in the two protocols. Different lines represent different participants. The distributions of relative phase across all participants and trials are plotted at the right margin. Participants mostly exhibited in-phase (0 deg) or an anti-phase (180 deg) relations between the cup and ball. **B)** Relative phase variability, computed as the circular variance of relative phase, represented the stability of the dynamics between the cup and ball. Mean relative phase variability across subjects is plotted against trials for the three pendulum conditions in the two protocols. The error ribbons represent $\pm 1$ standard error. **C)** Summary results of relative phase variability. Each data point in the box plot represents a participant average across 40 trials. Relative phase variability was similar between the two protocols, despite the increased uncertainty in the random protocol.

the stability of the kinematic relation between the cup and the ball. Relative phase variability was plotted over trials for the two protocols and pendulums in Fig 3B. There was no significant effect of pendulum length on relative phase variability ($F_{(2, 20)} = 2.64, p = 0.095$). Relative phase variability (log-transformed) was not significantly different between the random and blocked protocol, ($F_{(1, 10)} = 3.90$, $p = 0.076$) (Fig 3C). Overall, these results indicate that participants maintained stable relative phase, even when encountering uncertain dynamics.

**Prediction 1 - Relative phase is controlled during the preparation stage.** To address Prediction 1, we analyzed how relative phase changed during the preparation stage. Fig 4A shows a heatmap of relative phase from all trials and participants over the entire trial duration. Negative time indicates the preparation stage and positive time indicates the rhythmic stage. Relative phase converges systematically to either 0 or 180 deg by the start of the rhythmic stage for both protocols and all pendulum conditions.

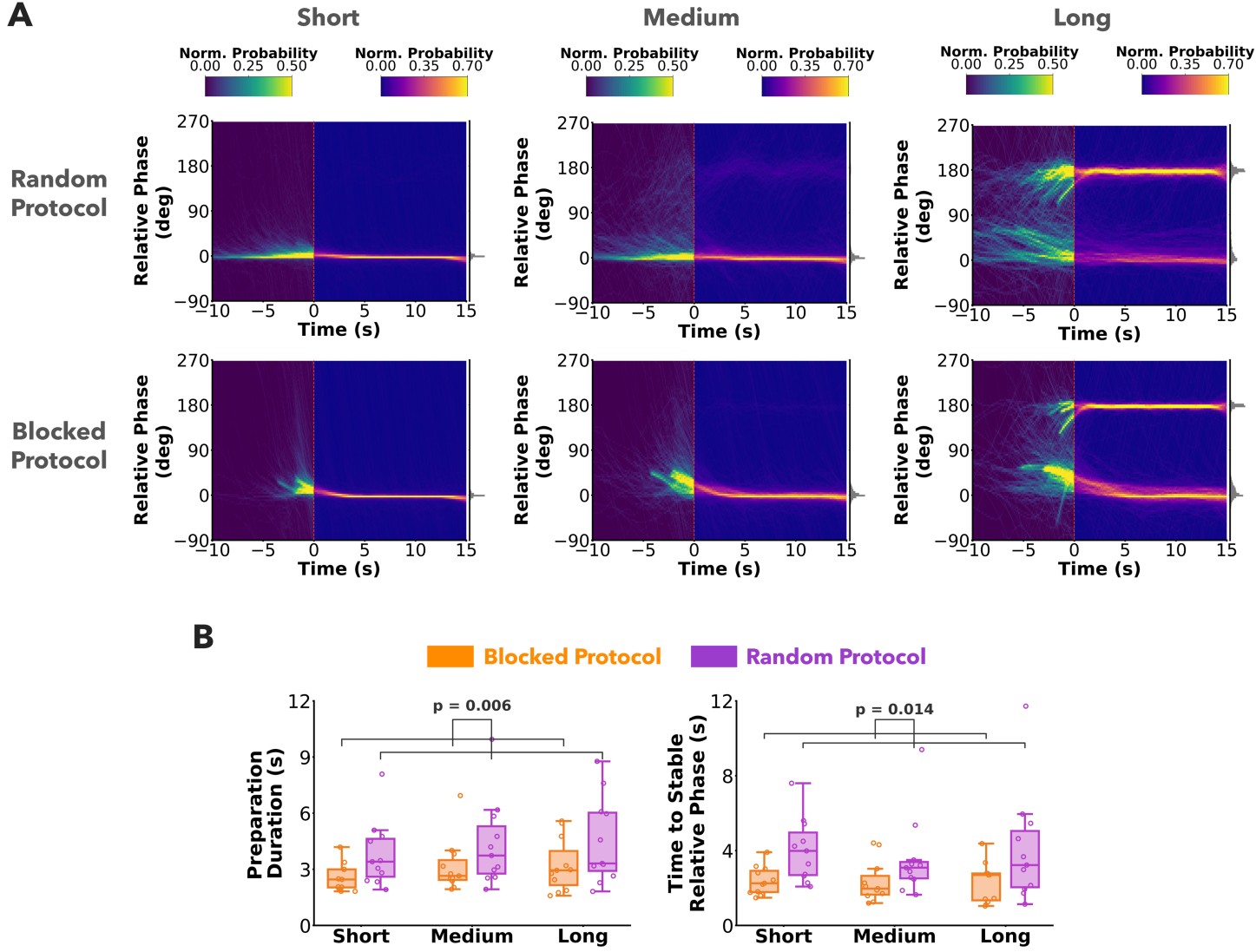

**Fig 4. Behavior in preparation stage.** A) Heatmaps of relative phase plotted against trial time from all participants and trials. Preparation stage is shown as negative time and rhythmic stage as positive time. The color scheme for the heatmap is mapped to the normalized probability and is different for the preparation and rhythmic stages. Relative phase converges within the preparation stage and remains fairly constant in the rhythmic stage in both random and blocked protocols. **B)** Summaries of duration of preparation and time taken to stabilize relative phase. Each data point in the box plot represents a participant average across trials. Participants utilized more preparation duration, and took more time to stabilize the relative phase in the random protocol in comparison to the blocked protocol.

This shows that participants used the preparation phase to move the cup-ball system to either a simple in-phase or anti-phase relation before executing the instructed rhythmic movements of the cup. Both preparation duration ($F(1, 10) = 11.97$, $p = 0.006$, $\eta^2 = 0.55$), and the time taken to stabilize relative phase ($F(1, 10) = 8.66$, $p = 0.014$, $\eta^2 = 0.46$) were significantly lower in the blocked protocol compared to the random protocol Fig 4B. These results gave support for Prediction 1 indicating that participants spent more time setting relative phase to desired values when the dynamics was uncertain.

**No trends in cup frequency and initial ball angle.** The cup frequency and initial ball angle for all participants in the two protocols are plotted over trials in Fig 5A and 5B. Participants could freely choose the cup frequency and the initial ball angle in each trial. As easily seen, individual participants adopted different cup frequencies that did not converge to

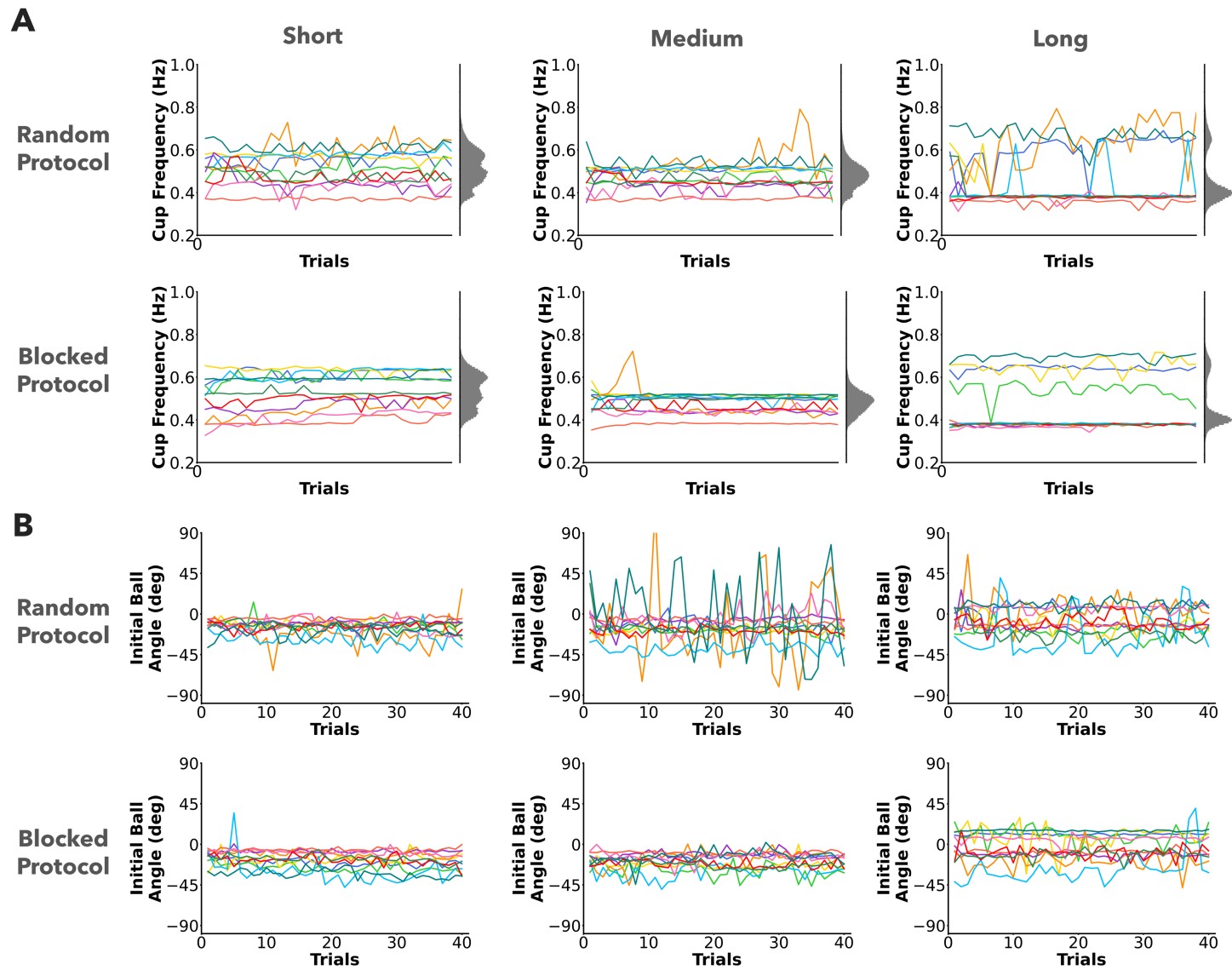

**Fig 5**. **Preparation and interaction strategies.** A) Cup frequency plotted over trials for the three pendulum conditions in the two protocols. Different colored lines represent different participants. Individual participants adopted different cup frequencies that did not converge to any particular common value over trials. The cup frequency distributions across participants and trials are shown in grey at the right margin. **B)** Initial ball angle plotted over trials for the three pendulum conditions in the two protocols. There were no observable trends in the initial ball angles or cup frequencies.

any particular common value over trials. The distributions of the cup frequency for the different pendulums, plotted in the margins of the timeseries, were statistically different for both protocols ($p < 0.001$) (grey histograms, Fig 5A). Similar to cup frequency, there were no uniform trends for initial ball angles over trials.

**Prediction 2 - Cup frequency and initial ball angle covary to ensure stability.** To test Prediction 2, we conducted forward simulations to determine how cup frequency and initial ball angle may have affected relative phase variability. We used a first-order impedance controller (Fig 6A) to model a participant moving the cup as described in the methods section.

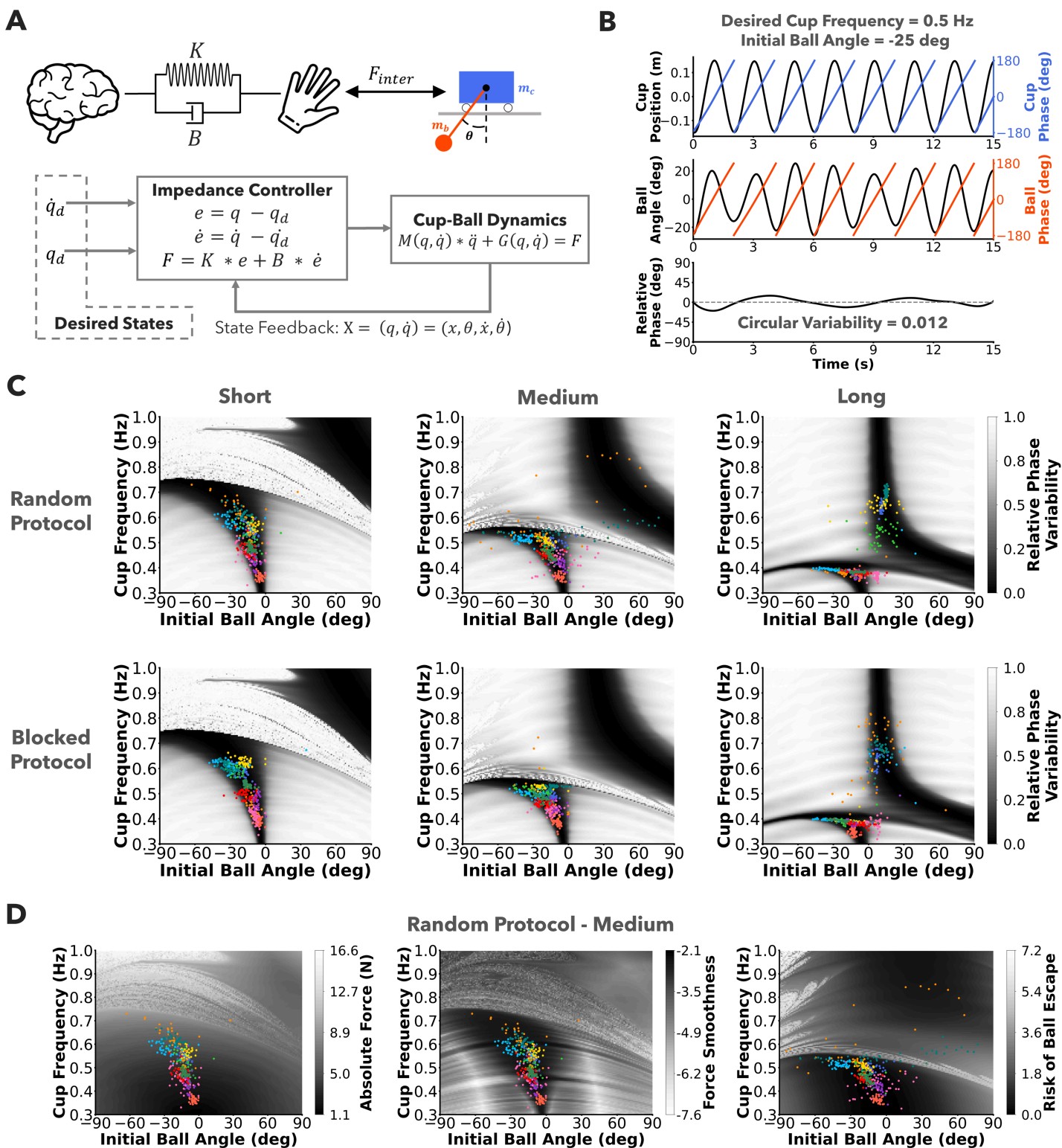

**Fig 6. Forward simulations to explain preparation and interaction choices. A)** A human-inspired controller with mechanical impedance was used to run forward simulations of the cart-pendulum model. The controller computed a force proportional to deviation from the desired states, related through the impedance gains $K$ and $B$. The impedance gains were held constant at $K = 40$ and $B = 70$. **B)** One forward simulation for initial ball angle of 25 *deg*

and cup frequency 0.5 *Hz*. The relative phase hovered close to 0 *deg* with a circular variability of 0.012. **C)** Heatmaps of relative phase variability for combinations of initial ball angles and cup frequencies plotted in grey shades in the background. Participant choices of initial ball angle and cup frequency from all trials are plotted as colored points in the foreground. Colors represent different participants. Participants' choices fell into the dark regions of the heatmap, showing that they nonlinearly covaried preparation and interaction strategies to maximize relative phase stability. **D)** Heatmaps of absolute force, smoothness of force, and risk of ball escape plotted for the medium pendulum condition. Participant choices of initial ball angle and cup frequency from the random protocol plotted in the foreground. Kullback-Leibler divergence between the distributions of participant data and the underlying heatmaps were lowest for relative phase variability, i.e., stability, (random-medium plot in **C**) compared to the three alternative costs. Relative phase variability captures covariation of preparation and interaction strategies better than absolute force or smoothness of force.

Cup position, ball angle, relative phase and relative phase variability (circular variability) are shown for a simulation run with desired cup frequency of 0.5 Hz and initial ball angle of -25 deg (Fig 6B). A heatmap of relative phase variability was plotted from simulation runs of all combinations of desired cup frequency and initial ball angle for each pendulum (Fig 6C). Data from all participants and trials was superimposed on the heatmaps for both protocols. The medium and long pendulums supported both in-phase and anti-phase solutions (two dark colored branches) in the preferred frequency range of 0.3 *Hz* to 0.7 *Hz* of the participants. Some participants adopted both in-phase and anti-phase stable solutions indicated by the same colored dots in the in-phase and the anti-phase branches, as was seen in the crossing over of lines from 0 *deg* to 180 *deg* in the relative phase plots in Fig 3A. The results gave support for Prediction 2 showing that participants flexibly and nonlinearly covaried cup frequency and initial ball angle to ensure low relative phase variability. This was seen irrespective of the experimental protocol.

To compare these analyses with alternative objective functions, similar heatmaps were created for mean absolute force, log dimensionless jerk of applied force, and risk of ball escape. The same participant data was overlaid on the heatmaps of the random protocol and the medium pendulum condition Fig 6D. KL-divergence was lower for relative phase variability (*Short* : 4.93, *Medium* : 5.21, *Long* : 5.29) compared to mean absolute force (*Short* : 5.93, *Medium* : 6.01, *Long* : 6.45), jerk of applied force (*Short* : 5.74, *Medium* : 5.98, *Long* : 6.52), and risk of ball loss (*Short* : 5.70, *Medium* : 5.95, *Long* : 6.47). Lower KL-divergence indicated that the simulated heatmap better explained the distribution of the data.

Hence, we concluded that the choice of cup frequencies and initial ball angle of the participants was best explained as minimizing relative phase variability. For brevity, we show heatmaps and report KL-divergences only for the random protocol and medium pendulum condition. The heatmaps and the corresponding KL-divergence results for other pendulum conditions are shown in the supplementary document S1 Text. These simulation results underscored the experimental results and supported Prediction 2.

**Prediction 3 - Mechanical impedance increased with increased uncertainty.** The grip force, proxy for mechanical impedance, was plotted against time for an example participant for all trials (Fig 7A, top row). The average grip force, shown by the bold lines, gradually increased during the preparation stage. During the rhythmic stage, the grip force reached higher values and remained higher in the random protocol. Fig 7A, bottom row shows the absolute value of the net force applied on the robotic handle plotted against time for the same participant for all trials. The applied force was overall at the same level for both stages and for both the random and blocked protocols.

The mean grip force and applied force across all trials and participants are summarized in Fig 7B. The grip forces in the random protocol were significantly higher than in the blocked protocol in the preparation stage ($F(1, 10) = 7.57$, $p = 0.020$, $\eta^2 = 0.43$) as well as in the rhythmic stage ($F(1, 10) = 9.06$, $p = 0.013, \eta^2 = 0.48$). We note that even when removing the one outlier participant, the grip forces in the preparation stage were still significantly higher in the random protocol.

In contrast, the applied forces in the random protocol were not significantly different from the blocked protocol in the preparation stage ($F(1, 10) = 0.53$, $p = 0.482$, $\eta^2 = 0.01$) or the rhythmic stage ($F(1, 10) = 0.47$, $p = 0.507$, $\eta^2 = 0.008$). Pendulum length significantly affected the applied forces in both the preparation stage ($F(2, 20) = 14.31$, $p < 0.001$,

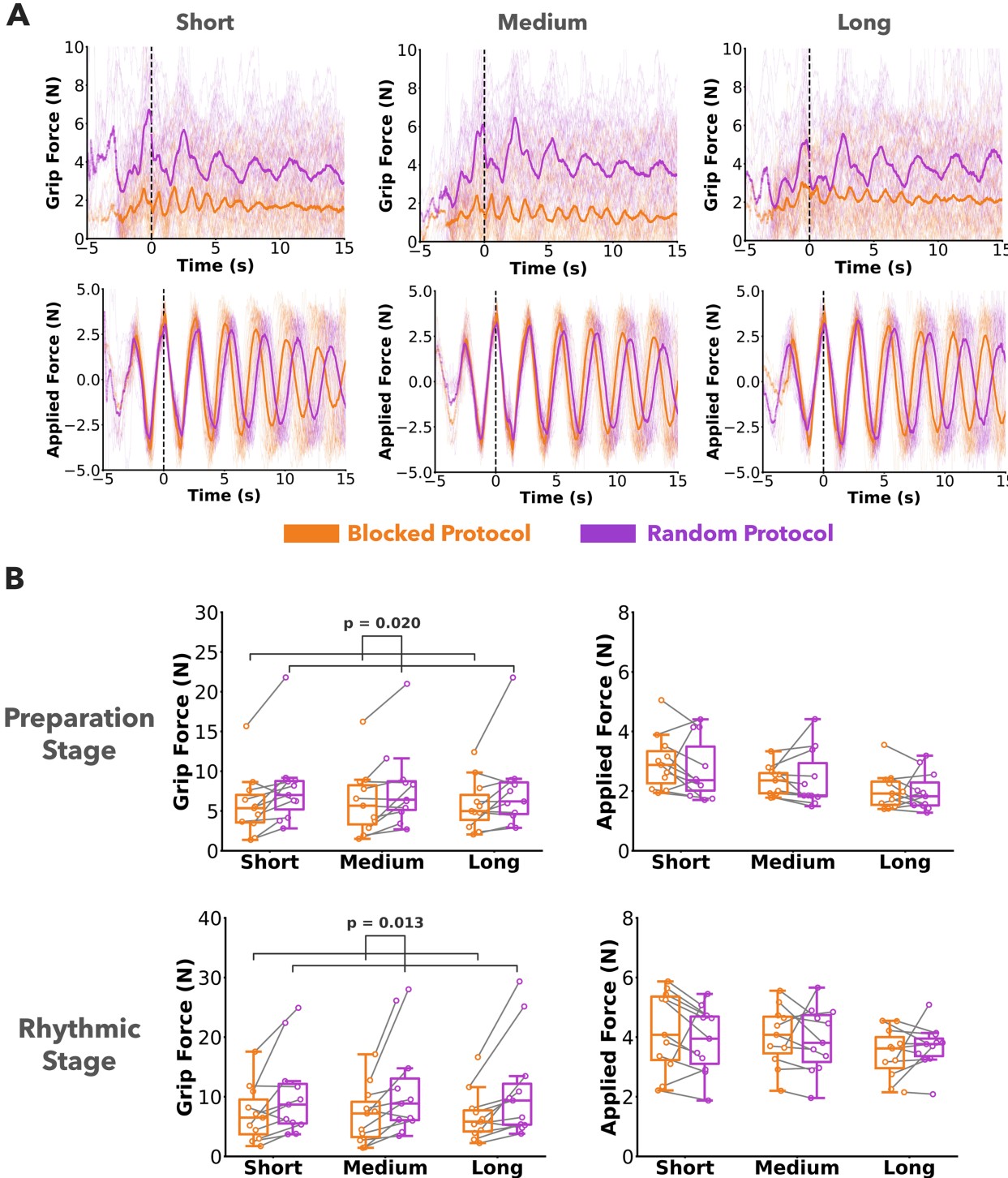

**Fig 7**. **Grip force and applied force.** A) Grip forces and applied forces from all trials of one representative participant are plotted as thin faint lines in the background. The mean grip force and applied force are plotted as thick solid lines. The participant increased the grip force during the preparation stage and reached a higher level in the rhythmic stage in the random protocol compared to the blocked protocol. In contrast, the applied forces were similar between the random and blocked protocols. **B)** Grip force and applied force from all trials for a participant plotted as individual points. Grey lines connect data points within the same participant. Grip forces were higher in the random protocol compared to the blocked protocol, both in the preparation and rhythmic stage. In contrast, applied forces were similar between the two protocols.

$\eta^2 = 0.68$)) and the rhythmic stage ($F(2, 20) = 8.29$, $p = 0.002$, $\eta^2 = 0.35$)). Overall, these results addressed Prediction 3 supporting that participants exhibited higher mechanical impedance when the uncertainty in the dynamics was increased. Importantly, the net forces were not simultaneously elevated.

**Modulation of impedance and forces in simulations match experimental results.** The states of the system from one Stochastic Optimal Open-loop simulation run for the random condition are shown in Fig 8A. The desired position and velocity of the cup during the rhythmic phase, and the ball angle at the beginning of the rhythmic phase are plotted in red. The numerically optimized pseudo-controls $\ddot{K}$ and $\ddot{F}$, and the corresponding impedance control variables $K$ and $F$ are also shown. Similar to the experimentally observed changes in grip force within a trial, the stiffness variable $K$ increased during the preparation stage and plateaued in the rhythmic stage. The stiffness and applied force were plotted for one representative simulation run for both the random and blocked conditions Fig 8B. The stiffness was higher in the simulated random protocol (higher uncertainty) compared to the blocked protocol (lower uncertainty). In contrast, the applied force was similar between the two protocols Fig 8B. The stiffness values averaged across all simulation runs were greater in the random protocol than in the blocked protocol (Fig 8C). However, the applied forces averaged across all simulation runs were similar in the random and blocked protocols (Fig 8C). Overall, the optimal modulation of stiffness and applied force agreed with the experimental data (Fig 7A and 7B).

**Simulated impedance accounts for preparation and interaction strategies.** Stiffness (Fig 8D) and absolute feedforward force (Fig 8E) averaged during the rhythmic phase were plotted as a function of initial ball angle and cup frequency for the simulated blocked and random protocols. Experimentally observed initial ball angles and cup frequencies from all participants in the random protocol were overlaid on the simulated heatmap. The choice of preparation and interaction frequencies indicated a preference for lower stiffness and lower applied force. Overall, these simulation results and the empirical data together suggest that minimizing end-point impedance as a plausible explanation for participants' choice of preparation and interaction strategies.

## Discussion

The results aligned with Prediction 1 that humans spend more time to prepare the object when dynamic uncertainty was higher. The preparation was primarily used to stabilize the relative phase between the cup and ball prior to rhythmic manipulation of the cup. Addressing Prediction 2, we observed that, irrespective of the experimental protocol, participants co-varied the initial ball states with cup frequency to ensure stable dynamics of the cup-ball system. Additionally, the stability of the dynamics took priority over effort, smoothness of forces, or risk of losing the ball, supporting the prediction from Prediction 2. Using grip force as a proxy for mechanical impedance of the arm, we observed that participants increased stiffness or mechanical impedance when the cup-ball parameters changed randomly from trial to trial, supporting the Prediction 3. Feedforward optimal control simulations confirmed the utility of higher impedance with increased uncertainty of the cup-ball system. Additionally, the simulations showed that participants choice of preparation of the ball and cup frequency may help to minimize mechanical impedance.

### Preparatory control strategies for complex object manipulation

The initial conditions of a nonlinear system can significantly influence its subsequent dynamical behavior [55]. Thus, the control of a nonlinear system can potentially be simplified by carefully setting the initial conditions of the system. For example, when spreading a table cloth, we first throw the cloth in the air such that it spreads out before gradually lowering it on the table. Preparatory strategies used by humans have been studied from a planning standpoint in less complex dynamic contexts like grasping an object [7,56]. Preparation has also been investigated within the body, such as anticipatory postural adjustments to account for external or internally generated perturbations [57–60]. These preparatory adjustments may be regarded as feedforward control strategies [61,62]. Our study focused on preparing the states of an object

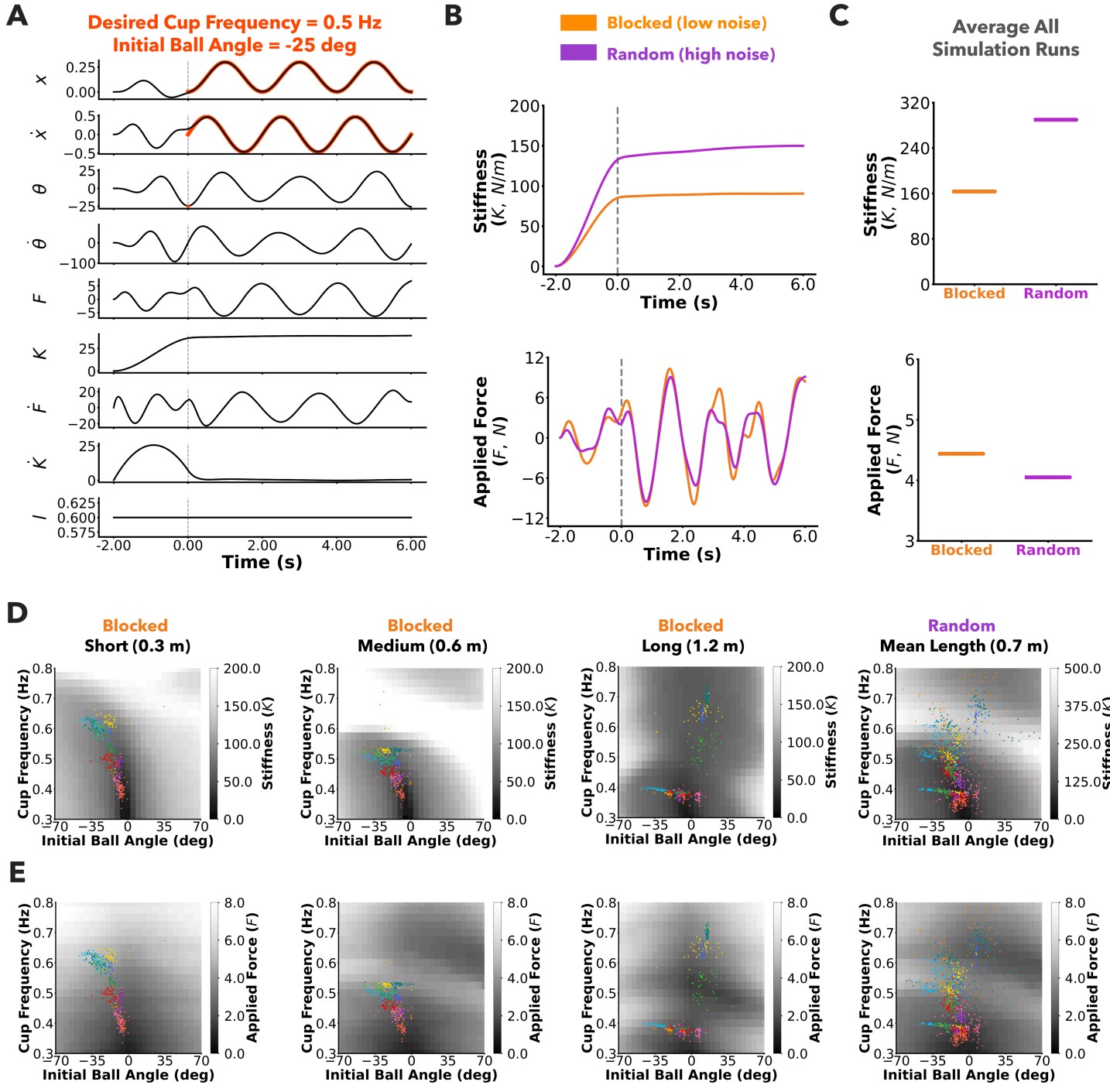

**Fig 8. Predictions of open-loop optimal control on mechanical impedance.** A) State space variables from one simulation run with the medium pendulum, and low noise simulating the blocked protocol, setting the desired initial ball angle to −25 *deg*, and cup frequency to 0.5 *Hz*. The desired cup trajectories and initial ball angle are highlighted in red. The stiffness *K* rises monotonously in the preparation stage and plateaus in the rhythmic stage. **B)** Stiffness and applied force plotted against time for the blocked (low noise) and random (high noise) conditions, for the same desired initial ball angle and cup frequency as in **A**. Predicted stiffness values are higher in the random protocol in comparison to the blocked protocol, similar to the experimentally reported grip forces. In contrast, applied forces are similar between the protocols. **C)** Average stiffness and applied force from all runs with combinations of desired initial ball angles and cup frequencies. Stiffness values are higher in the random protocol compared to the blocked

protocol. In contrast, the applied forces are of similar magnitude between the two protocols. **D)** Model predicted stiffness ($K$) for various combinations of initial ball angle and cup frequency plotted in the background. Participant choices from different trials from either the blocked protocol or random protocol are plotted in the foreground. Colors represent different participants. Participants covaried their choice of preparation and interaction to minimize mechanical impedance in both the random and the blocked protocol. **E)** Model predicted average absolute feedforward force ($F_{ff}$) for various combinations of initial ball angle and cup frequency plotted in the background. There are some regions of overlap between low average absolute forces and low stiffness.

and the interacting limb's mechanical impedance during a dynamic task and explores the feasibility of feedforward control strategies.

**Preparation and interaction strategies alone do not reveal systematic choices.** Participants in this study were instructed to 'play' with the cup and ball and start the rhythmic movement when they felt ready. However, our analysis did not reveal any systematic results when examining preparation or interaction strategies independently. The oscillation frequency of the cup in the rhythmic stage varied within and between participants. The bimodal distribution of frequencies into a lower mode and higher mode, corresponding respectively to in-phase and anti-phase relations between the cup and ball, was already reported in a prior study [24]. The preference for such simple one-to-one phase relations has also been observed between limbs in bimanual and bipedal coordination [63,64]. Similarly, the initial state of the ball at which subjects chose to start the rhythmic movements also varied within and between participants. This result contrasts with a prior study from our group with a similar task paradigm that found convergence of initial ball angles with practice [27]. The major difference of that study was that participants were enforced to oscillate the cup at the predetermined frequency of 0.6Hz. In contrast, in our study, participants could oscillate the cup at their preferred frequency. These observations indicated that the choice of preparation and frequency were most likely interdependent.

**Nonlinear covariation of preparation and interaction ensures stability of dynamics.** Further analysis revealed that preparation of the ball and oscillation frequency covaried flexibly to ensure stable dynamics of the cup and ball. To this end, stability was assessed as the variability of the relative phase between the cup and ball, similar to prior studies [24,27,28,65]. When initial ball angle and cup frequency were mapped into stability, the empirical choices of preparation and interaction frequency fell right into those areas of highest stability. These areas afforded two disjunct areas that accommodated the in-phase and anti-phase solutions that subjects chose. Thus, a few participants also switched between the in-phase and anti-phase solutions with the long pendulum.

**Stability accounts for behaviors better than smoothness and effort.** The subjects' choices were better explained by stability than magnitude of effort or smoothness of the applied force. Energetic and mechanical costs and smoothness have been used extensively to explain motor behavior within an individual, often supported by computational modeling [51,66–68]. There was indeed some overlap between maximizing stability and minimizing absolute effort or maximizing smoothness of applied force. It is therefore possible that humans optimize a weighted combination of the different variables. Further analysis is necessary to determine how maximizing stability could also lead to minimization of applied force. However, some participants chose slightly higher frequencies that required greater and less smoother forces as that still ensured stable dynamics. When manipulating complex objects, humans might prioritize objectives that go beyond effort and smoothness. A future study could test the robustness of this result and give participants more practice trials that engender fatigue so that effort expenditure and smoothness may become more relevant.

**Stability ensures predictability of underactuated dynamics.** This finding is consistent with several previous studies that demonstrated that humans seek predictability of the dynamics when interacting with objects of complex dynamics [18,24,27]. When the behavior of a system is stable, the dynamics of the underactuated degrees of freedom is more predictable [20,69]. For instance, in our task, when the variability of the relative phase between the cup position and ball angle is small, the ball motion is more predictable given the motion of the cup. It has been posited that increasing predictability affords a feedforward control strategy that is particularly useful when interacting with systems that can exhibit

chaotic dynamics [70]. Humans might prefer stable dynamics to obviate the need for computationally effortful elaborate internal models and/or error corrections with delayed sensory feedback [26].

### Role of mechanical impedance during complex object manipulation

We adopted a simple human-inspired impedance controller to generate the stability solution landscapes. This controller incorporated mechanical properties of the limb, like stiffness and damping, which are pivotal to maintain stable interactions with objects in our environment [29,40]. Humans have been shown to form an internal model of objects that they interact with [12,71]. But, forming an internal model can be challenging for unfamiliar objects that have nonlinear underactuated dynamics [26]. To account for uncertainty in the internal model stochastic optimization predicts modulation of mechanical impedance [72].

For the forward simulations, we adopted constant stiffness and damping gains that sufficiently minimized deviations from the desired sinusoidal target trajectories of the cup, similar to previous work [24,27]. Despite the simplicity, the stability landscapes sufficiently matched participants' control strategies in our experiment. Although assuming constant levels of limb impedance is quite common, it is important to acknowledge that humans carefully modulate mechanical impedance of the limbs [73–75]. Indeed, the grip force measurements that represented mechanical impedance of the arm varied intricately within a trial. One potential reason for these variations could be the coupling between the load force and grip force, i.e., impedance may increase in response to an increase in the reaction force on the hand from the cup [76,77]. This is consistent with the observation that increase in impedance corresponded to an increase in amplitude of the applied force during the preparation stage. Another potential explanation is that higher impedance could attenuate the unpredictable underactuated dynamics, i.e., the forces from the ball on the cup.

**Manipulating mechanical impedance to enhance sensory perception.** An alternative viewpoint is that the stiffness of the arm is modulated to enhance perception of the states of the environment [78]. Modulation of stiffness to extract more information was shown in a dynamic tracking task, where participants increased the stiffness of their wrist when haptic force feedback information about the tracked target was more noisy [79]. Such a strategy is closely related to the concept of 'active perception' where movements are purposefully directed to enhance perception, i.e., to extract more information about the object that we are interacting with [80,81]. Active perception via physical interaction also has received significant attention in robotics research for navigation and interaction in unknown environments [82–84]. It is therefore possible that participants in our study increased their end-point stiffness to better perceive the ball dynamics. Future experiment can be designed to systematically test whether increasing impedance enhances haptic sensing of the underactuated dynamics.

**Control framework captures the modulation of mechanical impedance.** To explain the results on mechanical impedance, we used the stochastic optimal open-loop control framework. The stochastic open-loop optimal control framework solves a deterministic optimal control problem for the optimal feedforward force and impedance [54]. The use of this framework was justified by prior research showing that impedance can be planned and implemented in a feedforward fashion to ensure robustness to neuromotor noise and mechanical perturbations [30,33,36]. Stochastic optimal open-loop control simulations have been used to explain force and impedance modulation reported by seminal literature on sensorimotor control [38,39,85].

In accordance with the changing magnitude of grip force within a trial, the simulations also allowed impedance values to vary within a trial. In the random protocol, a higher impedance resulted from the simulations to control the variability in the cup states. The impedance gradually increased in the preparation stage, both in the experiment and in the simulations. The gradual increase in mechanical impedance during the initial stage was due to the costs imposed on smoothness of changes in impedance and higher impedance being unnecessary in the preparation stage. Since we were primarily interested in capturing the differences between the two protocols, we discounted the modulation of the impedance within a trial in the rhythmic stage.

Finer modulation of mechanical impedance within a trial could have been enabled by relaxing the costs on time derivatives of mechanical impedance or incorporating feedback via state-estimation methods. Additionally, it is possible that visual and haptic feedback of the dynamics of the ball were also involved in determining the impedance of the arm. The feedforward impedance would then also minimize the fluctuations of the cup states observed in the simulations. Further experiments and simulations including feedback control are required to comprehensively understand the effect of underactuated dynamics on real-time modulation of mechanical impedance.

**Impedance is optimized via choice of preparation and interaction strategies.** The choice of interaction frequency and ball preparation also influenced the mechanical impedance. Although feedforward planning of impedance is likely more efficient than feedback control of errors, it is not energy optimal. Co-contracting pairs of agonist-antagonist muscles to increase joint stiffness trades energy for lower variability and higher stability [72,86]. When there are multiple solutions to a task, it would be energetically more efficient to choose a solution that requires lower mechanical impedance. Coincidentally, the participants' choice of preparation and interaction strategies also minimized mechanical impedance, i.e., the regions of low impedance derived from the optimal control simulations aligned with regions of greater stability of the ball dynamics, derived from the forward simulations. Our results therefore also point to a physiological rationale for humans seek stable dynamics: it may reduce mechanical impedance that also reduces effort expenditure. This leaves open the question of causality, whether stable and predictable dynamics facilitates lower mechanical impedance or whether impedance leads to stable dynamics- a question left for future research.

## Preparation stage facilitates preparation of object and impedance

Participants in the experiment used the preparation stage to set up states of the object and mechanical impedance. They gradually stabilized the relative phase between the cup and the ball over the course of the preparation interval. Consequentially, the ball angle at the end of preparation was determined according to the frequency adopted in the preparation stage. When the pendulum length was uncertain, participants nevertheless achieved a similar degree of stability, but preparation times were longer. Besides, the mechanical impedance of the arm was increased over the preparation stage, which was also reproduced in the simulations. Thus, preparation was not only limited to the kinematic states of the object, but also included mechanical impedance. Overall, the preparatory strategies allowed a smooth transition into the rhythmic stage and helped ensure stability of dynamics in the remainder of the trial. It is worthwhile to consider that the preparatory stage may have served to explore and reduce the dynamic uncertainty by identifying the pendulum length parameter. However, with such a process, it could be expected that identification of system parameters should lead to a change in control strategy. It is an open question whether identification of system parameters is necessary during such interactions and how exploration of dynamics helps facilitate system identification.

## Conclusions

Humans are exceptional at interacting with objects that have complex uncertain dynamics. Our findings throw light on feedforward motor control strategies used by humans when interacting with objects that have nonlinear and underactuated dynamics. Preparing an object to a favorable initial configuration and tuning the frequency of the applied force are two different fundamental contributions that covaried and thereby ensured stable dynamics. Mechanical impedance can be optimally prepared and modulated to deal with uncertain underactuated dynamics. We believe our findings are significant as they can potentially aid in the development of control strategies for robotic manipulation, where interaction with flexible objects is still a holy grail. Furthermore, insights may inform rehabilitation strategies for patients with neurological injuries as interactions with objects [87] are inherent to almost all activities of daily living.

## Supporting information

**S1 Text.** This first section in the supplementary information document reports forward simulation results like Fig 6D for all the pendulum conditions (Short, Medium and Long) and variables (Absolute Force, Force Smoothness, and Risk of Ball Escape) in **Fig AA**. The corresponding KL-Divergence results are also summarized in **Fig AB**. The next section reports in detail the cost and state weighting matrices that were used in the Stochastic Open-loop Optimal Control simulations. (PDF)

## Author contributions

**Conceptualization:** Rakshith Lokesh, Dagmar Sternad.

**Data curation:** Rakshith Lokesh, Dagmar Sternad.

**Formal analysis:** Rakshith Lokesh, Dagmar Sternad.

**Funding acquisition:** Dagmar Sternad.

**Investigation:** Rakshith Lokesh, Dagmar Sternad.

**Methodology:** Rakshith Lokesh, Dagmar Sternad.

**Project administration:** Rakshith Lokesh, Dagmar Sternad.

**Software:** Rakshith Lokesh.

**Supervision:** Dagmar Sternad.

**Validation:** Rakshith Lokesh, Dagmar Sternad.

**Visualization:** Rakshith Lokesh, Dagmar Sternad.

**Writing – original draft:** Rakshith Lokesh, Dagmar Sternad.

**Writing – review & editing:** Rakshith Lokesh, Dagmar Sternad.

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
