## [Decision Letter · Decision Letter 0]

13 Aug 2025

PCOMPBIOL-D-25-01193

Dynamical stability and mechanical impedance are optimized when manipulating uncertain dynamically complex objects

PLOS Computational Biology

Dear Dr. Lokesh,

Thank you for submitting your manuscript to PLOS Computational Biology. After careful consideration, we feel that it has merit but does not fully meet PLOS Computational Biology's publication criteria as it currently stands. Therefore, we invite you to submit a revised version of the manuscript that addresses the points raised during the review process.

Please submit your revised manuscript within 60 days Oct 13 2025 11:59PM. If you will need more time than this to complete your revisions, please reply to this message or contact the journal office at ploscompbiol@plos.org. Please include the following items when submitting your revised manuscript:

We look forward to receiving your revised manuscript.

Kind regards,

Bastien Berret

Guest Editor

PLOS Computational Biology

Daniele Marinazzo

Section Editor

PLOS Computational Biology

**Additional Editor Comments:**

As you will see, two reviewers had minor concerns but one reviewer raised more major concerns. To address these issues, your revised manuscript should provide substantial clarification on the following points: (1) the novelty and rationale of this work compared to prior studies using a similar paradigm, (2) the hypothesis-driven approach that underlies the study, and (3) some key choices of the modeling and how it relates to the behavioral task. Please note that all relevant data should also be shared, if possible, according to the policies of the journal.

**Journal Requirements:**

At this stage, the following Authors/Authors require contributions: Rakshith Lokesh, and Dagmar Sternad. Please ensure that the full contributions of each author are acknowledged in the "Add/Edit/Remove Authors" section of our submission form.

Potential Copyright Issues:

i) Figures 1B, and 5A. Please confirm whether you drew the images / clip-art within the figure panels by hand. If you did not draw the images, please provide (a) a link to the source of the images or icons and their license / terms of use; or (b) written permission from the copyright holder to publish the images or icons under our CC BY 4.0 license. Alternatively, you may replace the images with open source alternatives. See these open source resources you may use to replace images / clip-art:

5) We note that your Data Availability Statement is currently as follows: "All relevant data are within the manuscript and its Supporting Information files". Please confirm at this time whether or not your submission contains all raw data required to replicate the results of your study. Authors must share the “minimal data set” for their submission. PLOS defines the minimal data set to consist of the data required to replicate all study findings reported in the article, as well as related metadata and methods (https://journals.plos.org/plosone/s/data-availability#loc-minimal-data-set-definition).

2) If any authors received a salary from any of your funders, please state which authors and which funders..

7) Kindly revise your competing statement to align with the journal's style guidelines: 'The authors declare that there are no competing interests.'

**Reviewers' comments:**

Reviewer's Responses to Questions

**Comments to the Authors:**

Reviewer #1: This study used a pendulum cart haptic system to investigate how human participants control an object with an internal degree of freedom. By looking at the cup’s movement frequency and the ball’s initial angle during the preparation stage, they examined how people change their behavior in the preparation stage depending on the uncertainty in the pendulum’s length, which changed its dynamics. They found that participants’ preparation behavior was best explained by a model that minimized the variability in the relative phase between the cup and the ball, i.e., by prioritizing predictability in the task dynamics. The Introduction is thorough and appropriate, the Results are well described, the simulations are informative, and the Discussion and Conclusion reflect the study’s findings. I have only minor suggestions and points that require clarification.

According to line 187 “the rhythmic stage started when the cup was first moved towards the box on the right side”. But looking at Fig. 1C, it seems like the trial starts when it hits the left box. Is there a mistake in the text? Also, please state explicitly how the initial ball angle was calculated. Was it the angle when the cup first entered the box? Or was it its peak angle when inside the box on the first occasion?

I would use the same color scheme in Fig. 3B as in Fig. 2.

In Fig. 3C, please make the line styles different for each pendulum length and make the lines thicker. I found it difficult to discern which plot corresponded to which pendulum length.

The authors used a three-way ANOVA to test the difference in the relative phase variability between the random and blocked protocols, but upon examining Fig. 3D, I don’t think the data is normally distributed to warrant an ANOVA. Did the authors check that normality was not violated? Perhaps they could try taking a log of the relative phase variability and run their tests again to see if normality is preserved for further analysis with an ANOVA.

For Fig. 4B, were no post-hoc tests conducted? I would define the time taken to stabilize the relative phase when it is first introduced in the Results on line 283 instead of only in the Methods. Could the authors justify their selection of the 0.1 threshold? How was this determined?

Concerning Fig. 4B, the number of cycles and preparation duration appear redundant, I would take just the latter since this is the most intuitive. Preparation duration and time to stabilize relative phase are different, but there is no further analysis beyond the three-way ANOVA. In the random protocol, the time to stabilize with the medium pendulum appears to be somewhat shorter than with a short and long pendulum, is this difference significant? Please add more details here, or if the conclusion is relatively the same, then I would replace all three metrics with just preparation duration.

Line 329: “loosing the ball”  “losing the ball”

Is the plot in Fig. 5D available for all conditions? I didn’t see Supplementary Materials showing these figures. I would add them either in the main manuscript or as Supplementary.

Was the Kullback-Leibler divergence computed for each condition and then averaged across all conditions? Please specify.

There are several black splotches in the simulations of Fig. 7D, why did the stiffness produced by the controller sometimes drop close to zero in some cases, e.g., medium pendulum with cup frequency around 0.7 Hz and initial ball angle around 20 degrees?

I would recommend adding another subfigure in Fig. 7 like 7D but with the mean applied force just to see how stable it is for different conditions.

The local fluctuations in grip force (which weren’t modelled) could be an artefact of the coupling between grip and load (applied) forces and may not be indicative of a voluntary impedance control strategy (Hadjiosif and Smith 2015). This could be added to the Discussion.

While the relative phase variability matches best with the experimental data, this does not preclude the possibility of a weighted combination of the metrics, e.g., relative phase variability and absolute force. I would add a sentence in the Discussion describing this limitation.

References

Hadjiosif AM, Smith MA. Flexible Control of Safety Margins for Action Based on Environmental Variability. J Neurosci 35: 9106–9121, 2015.

Reviewer #2: This article investigated an interesting question: what are the human motor control strategies during the dynamic manipulation of complex and uncertain objects. They used a simplified cart-pendulum model to simulate a nonlinear and under-actuated physical system, introducing system uncertainty by varying the length of the pendulum ball. The study found that humans adopt different preparatory strategies under higher uncertainty and prioritize dynamic stability for their motor control, increasing muscle stiffness to cope with uncertain dynamics. Feedforward simulations using an impedance controller and stochastic open-loop optimal control support these findings.

The paper was interesting and enjoyable to read. I have the following comments to improve the paper and clarify some statements.

Major comments:

1. At Line 23 of Abstract, it states that “further revealing that participants selected preparation and interaction frequencies to minimize mechanical impedance”. It infers that the main goal for the motor control is to minimize mechanical impedance that is related to the effort level, but previously from Hypothesis 3 it states that the dynamics stability was prioritized. Please make sure that the statements are consistent.

2. At Lines 359-360, it states that “The average grip force, shown by the bold lines, gradually increased during the preparation stage.” This is interesting as the dynamics are unchanged within the trial. The increase of the impedance is probably due to different requirements in the rhythmic stage. The applied force seems also having an increase in rhythmic stage compared to preparation stage from Fig. 6B. Please add discussions on the different behaviours between the two stages and potential reasons.

3. In Fig. 3C, the medium and long colours are close, which are not very easy to distinguish. Besides, please describe Fig. 3C that the meaning of the solid line in figure caption. The long pendulum results (dark blue line) look consistently higher than the other two, however there is no significance detected from statistics. The results in Figs. 3C& D look not very consistent, please add some additional descriptions to clarify this. Besides, from 3B, it looks also that the long pendulum has more variance compared to the short and medium, and more switches between the 0 deg and 180 deg, especially for the random protocol. How would this reflect the task difficulty level or influences stability control?

4. In eq. (5), the double derivative of impedance and force are chosen as control input, however as human impedance modulation may relatively slow dynamics, controlling the double derivative of the impedance may impose constraints to the control inputs. This would turn it into a constrained optimal control problem. Please clarify whether these control constraints have been considered in the current modelling framework. Besides, could you please share some information on how using the (K,F) or their first derivative as control input would change the results?

5. At Lines 915-916, “An initial guess for F was obtained using the impedance controller defined in Eq. 10 using a small gain K to improve the convergence speed of the numerical optimization.” The F here is the feedforward force control, while the force in eq. 10 is more for feedback force control. Besides, eq. (10) and eq. (12) share the same symbol F but they have different meanings, please use different symbols for these two, otherwise there will be two feedback control terms in eq. (12).

6. At Lines 574-575, the authors claim that humans prioritize objectives going beyond effort and smoothness, however I am entirely convinced by this claim. As they mentioned that people fatigue may change the results, I was wondering whether authors have looked at how the gripping force changes along trials and whether the Kullback-Leibler divergence values are consistent between the beginning and finishing trials. From Fig. 3C, there is a slight increase trend in the short and medium pendulum between trials 30 and 40. Some further analysis would be necessary to support this claim.

Besides, the authors mainly consider the force smoothness in Fig. 5D, but the movement smoothness is important to consider, so it would be beneficial to analyse the movement smoothness.

Minor:

1. At Line 195, it told that there are “210 trials” in total for each protocol, however Fig. 1D shows that there are “120 trials”, please make sure that the number of trials is correctly reported. Similar issue exists at Line 196.

2. At Lines 237-238, anti-phase relations are mentioned without further explanations. Please give some descriptions on whether anti-phase relations are good or bad to the stability.

3. In Fig. 5D, the force smoothness is flipped for some reasons, but it would be better to make it consistent with others.

4. At Lines 423 & 878, the dimension of u should be 2 instead of 3.

5. At Line 914, B is not used in the formulation. As the control inputs are the double derivative, should these initial guesses be the double derivative of K and F correspondingly?

6. In eq. (9), the lower bound of the integral should be added.

7. In eq. (8), please define clearly what is the \theta_{rim}.

8. At Line 749-750, it states that “In the preparation stage, participants were encouraged to explore the cup and ball dynamics by ’jiggling’ the cup between the left block (−0.15 m) and the midpoint at 0 m.”, However, in Fig. 1B, the midpoint position is not provided in the visual feedback, how does the participant know the midpoint position? Besides, why space between left block and the midpoint is selected in the preparation stage? Would this influence the results of preparation if subjects have different handedness?

9. At Lines 751-753, it states that “When participants were ready to start the rhythmic stage, they moved the cup towards Box B, and then continuing with rhythmic back and forth movements between the two target boxes.” Please add more specifically what information has been provided to the participants: are they aware of the trial duration or whether they are informed of any rewards or penalties on the movement control? This would be useful to understand their motion and impedance behaviour.

10. At Line 706, please add the mean and standard deviation of participants’ age.

11. At Lines 736-737, “an admittance-controlled robotic manipulandum (HapticMaster, Motekforce, Netherlands, Fig. 1B).” Is the admittance-control used in the experiment? The admittance-control may changes the dynamics of the system. If I understand correctly, the dynamics eq. (6) was implemented in the handle movement, please clarify this.

12. In Fig. 7D, the simulation results in the low stiffness area of the long pendulum blocked plot seems not very continuous compared to short and medium plots, can you please give some explanations for that?

13. At Lines 489-491, “Stiffness and applied force… simulated random protocol (Fig. 7D)” However, only stiffness has been displayed in Fig. 7D.

14. For the cost function between Lines 437 and 438, what are the desired values used for the stiffness and force as well as their derivative in the desired states?

Reviewer #3: This paper, overall, presents several interesting and potentially important ideas. I have quite a few specific (line-by-line) comments below, but they generally can be lumped under 3 general categories, 1 fairly straight-forward, 1 likely manageable, but one potentially challenging:

On the straight-forward side, there is a lot of duplication across “Results” and “Methods”. This is distracting and unnecessary. But it can also be relatively easily resolved by more clearly separating “Results” from “Methods” – in whichever order the authors feel it is best to present these.

“Manageable” is the conceptual issue of presenting this work as testing multiple “hypotheses” that are really not “hypotheses” but instead “predictions” and “explorations” of the experimental manipulations and results. Again, however, this should be somewhat easy to resolve in re-writing the narrative.

Most challenging is the question about the “So What?” question. It is very clear that a lot of work was done – experimentally, analytically, computationally, etc. however, it is not at all clear in the current narrative “To what end?” – i.e., what new important insights this work has to offer beyond a now extensive body of work already published (mostly, but not all, by this group) on this general ball-cup paradigm and these general questions surrounding “dynamic stability”, “impedance” and “what do people control and how?” when performing these tasks.

The specific comments below articulate the reasons for this assessment and offer (I hope) a roadmap…

ABSTRACT:

As currently written, it is not clear if the “hypotheses” came before the experiments and data… Or the experiments and data came first. As written, the experiments are described first. So it’s not clear that these were “hypotheses” or merely “findings”.

LL. 23-25: The general overall conclusion does not appear substantively or substantially different than multiple prior papers by the senior author’s group. The group has done very strong work over more than a decade now on this paradigm and the general question (L. 5) of “It remains unclear what motor control strategies ensure stability…” etc. The current abstract gives little-to-no indication what truly new, or novel, or important but as-yet-unanswered questions this present work offers.

AUTHOR SUMMARY:

See comments above regarding the abstract. Same issues here. As written, it’s a nice description of what was done and what the results were. But that is about all. Does not answer the “So What?” question.

INTRODUCTION:

LL. 46-65: A nice overview… But seems somewhat dated given the extended prior work on manipulating “dynamically complex” objects, by this group and others, over the past 2+ decades now.

LL. 65-66: 50-52: The starting statement that “To date, only a few studies have explored…” is incommensurate with (1) the number of studies then described that have addressed this question, and the fact that those studies date back to at least… looks like 2002 is the oldest one cited here? So more than 2+ decades of work on this topic area now?

LL. 89-90: Unclear. Sensitivity to initial conditions (i.e., ‘the butterfly effect’) is typically a hallmark of chaotic nonlinear systems… But many systems are nonlinear and not “chaotic”. To my knowledge, the ball-cart system studied here, while certainly “nonlinear” is not “chaotic” – or at least is not studied in any chaotic region of its dynamics (even if such regions exist mathematically), as people in fact do maintain stability of their overall movements in these experiments, as do the model simulations.

LL. 100-101: While this paragraph (LL. 89-100) summarizes prior work from this group and while “it remains unclear…” But the Intro does not articulate how or why this question is important, beyond being the next incremental step in disentangling this specific paradigm that the senior author has been working on for (at least) 15+ years now.

LL. 102-118: The concept of regulating mechanical impedance has been around 4+ decades now (Hogan, 1984 etc.). And the senior author has at least 4 prior papers on this ball-cup paradigm that incorporate at least some aspects of mechanical impedance in their models (Maurice, 2018; Nayeem, 2021; Razavian, 2023; & Bazzi, 2024). While this narrative summarizes key prior findings, it does not (by itself) provide a strong basis for the present study.

LL. 110-111: “… plan the impedance in a feedback-based manner, eliminating the need for feedback-based corrections…” doesn’t seem to make sense (“feedback” used twice). I suspect one of the “feedback” was intended to be “feed-forward”?

LL. 118-122: Again (see comment on LL. 102-118) – The argument as presented that “These models did not allow… Therefore, it is crucial to explore…” does not articulate *why* “it is crucial to explore…” – especially given the extensive prior work on these ideas and even within this specific ball-cup paradigm.

LL. 126-127: The Intro describes how this new work builds on previous work from this group… But does not articulate how or why the new ideas and/or results presented here are more than an incremental improvement over that (now extensive) prior work.

LL. 135-143: While “hypotheses” are presented, it is not at all clear where they came from or how they emerged from prior work. Furthermore (and perhaps more importantly), they are not so much “hypotheses” as (experimental) “predictions” – i.e., they predict the behavior the authors expected to observe in their experiments, but they do not address underlying physiological (or other) mechanisms that would (or might) be responsible for those observations.

RESULTS:

LL. 145-193: Most of this description is “Methods” not “Results”. And most of it describes the same methods used in many prior papers on this paradigm by this group. It is not made at all clear which aspects of these tasks were ‘same as used in prior studies’ versus which were unique and new to this study.

LL. 189-191: It is not at all clear here how using different pendulum lengths manipulates "uncertainty in the cup-ball dynamics"... But this is a sequencing issue. The manipulating of uncertainty part is not about the different pendulum lengths themselves, but rather the random vs. blocked protocols… But those are not described until the next paragraph… (so just some gentle re-ordering of the description here I think…)

LL. 161-167, Fig. 1D: It seems the “Random Protocol” was conducted (for all participants) on Day 1, while the “Blocked Protocol” on Day 2.

LL. 204-205: However, the statement that “order… was counterbalanced” contradicts what is shown in Fig. 1D… Or Fig. 1D is not consistent with how the experiment was actually performed. One or the other or both should need to be fixed to clarify.

Pgs. 4-7: In general, most all of what is presented on these ~3½ pages is “Methods” not “Results”. PLoS Comp Bio does allow authors to present their work as Intro-Methods-Results… If the authors feel this “Methods first” approach is best here, the manuscript should be restructured accordingly. If they believe the “Results first” approach is more suitable, then most all of this “Methods” description should be moved to “Methods” accordingly.

LL. 215-228: This is a lengthy description to end with the conclusion that there’s “nothing to see here”…? It seems like 1 or 2 brief sentences would suffice, rather than two whole paragraphs.

LL. 229-261 & Fig. 3: It is not made clear if these results represent anything new or different from prior experiments using this same paradigm. If this and the prior section (Figs. 2 & 3) are more “quality check” (?) (or “confirmatory” – the paradigm ‘worked’?) than new “Results” then consider moving to a Supplement and shorten the descriptions in the main paper to a sentence or two (e.g., “We confirmed that… (see Supplement ##)…” etc.)

LL. 262-286 & Fig. 4: Not clear that the prior comment might not also apply here as well….

L. 263 vs. L. 285: More importantly, L. 263 states “Given these first seemingly unsystematic results, we determined…” So these analyses were conducted after obtaining and observing the experimental finding. Yet, L. 285 states “These results showed support for Hypothesis 1…” If both statements are true, this would seem to be a clear case of the (usually unethical) practice of “HARKing” (https://en.wikipedia.org/wiki/HARKing). Instead, I believe this was not a “test” of a “hypothesis” but instead a “prediction” based on preliminary observation of the date (as L. 263 indicates). The result itself is fine… But just should not be framed as testing a “Hypothesis” if that is not what was done.

LL. 288-291: Same problem. “Having seen…, we examined….” (LL. 288-290) clearly frames this as an exploratory (not apriori hypothesis-driven) analysis. That is contrary to “To test Hypothesis 2, …” (LL. 290-291). Exploratory analyses are fine, and in fact often necessary, especially in computational work… But should be framed and presented as such.

LL. 291-301: Too much (and too detailed) “Methods” description for a “Results” section…

LL. 327-347: Again, same issue as above – The simulations offer post-hoc insights into the already-obtained and observed experimental findings (which is fine!), but they do not test some a priori “Hyopthesis”…

This section, In General: The simulations themselves seem fine, but as presented, were clearly devised a posteriori - *after* the experimental results had been obtained and assessed in some sense. Such simulations help “explain” (a posteriori) and offer additional context to the experimental results obtained… But that is very different than testing some (a priori) “Hypothesis” (e.g., L. 324 & L. 327, etc.).

LL. 339-342: Numerically, yes, the K-L divergences for minimum relative phase are “smaller” than for the other criteria (shown in Fig. 5D). However, (1) Fig. 5D shows the experimental data lying also at relatively (close to?) minimal values of those alternative criteria [abs. force, force smoothness, risk of escape], and (2) context for the numerical K-L divergence values is missing: i.e., what does a K-L divergence of 4.93 “mean” – physically and/or functionally etc.? How physically and/or functionally meaningful are the differences between, say, 4.93, 5.93, 5.74, and 5.70 (taking the “short” values just as examples)? Though the minimizing relative phase variability criteria seems “slightly better” than the others tested here, based on this K-L divergence metric, the data themselves (Fig. 5D) don’t seem to unequivocally rule out the other criteria, or some combination of these various criteria. This is a long-standing and somewhat common issue in motor control (e.g., https://doi.org/10.1006/jmps.2000.1295).

LL. 349-357: Again, more “prediction” than “hypothesis” (for same reasons stated above)…

Fig. 6: Plots in Fig. 6A don’t appear particularly “representative” – The separation between Blocked and Random appears much larger for this particular subject than the group data (Fig. 6B) might suggest.

LL. 379-380: The bottom part (Rhythmic Stage) of Fig. 6B suggests at least 2 outliers…?? As the sample size for these experiments (N=12) was not especially “large” (though likely adequate) and these comparisons seem a bit close-to-marginal, it would be helpful for others trying to interpret the true strength of these findings if calculations of effect sizes were included also. I’d recommend doing this for all of the analyses performed, to the extent feasible.

LL. 393-408: The description given here is (1) “Methods” not “Results”… But more importantly (2) rather confusing. Was the pendulum length varied as “Gaussian noise” (L. 398), or as “a discrete random variable…” with 3 distinct values (the distribution of which might be random, but would certainly not be “Gaussian”). Likewise, LL. 403-404 states that to simulate the blocked protocols, pendulum lengths were set to 3 discrete values “with a standard deviation…” But in the experiment, the (virtual) pendulum lengths were fixed – NO variation – Yes? The experimental variation was in how the 3 pendulum lengths were presented trial-to-trial (i.e., blocked vs. random), but NOT in the lengths of the pendulum itself – Yes? Either something in the descriptions of either the experiment, the models, or both is missing, or these simulations simulated something quite different from the experiment that was conducted.

LL. 393-454: Again – All of this is “Methods” – Not “Results” (see prior comments).

LL. 423-424: Still no explanation as to why Gaussian noise was added to the pendulum length – which (still) seems clearly contrary to the experiment conducted!...??

LL. 494-496: This statement is an interpretation of the results – Not a statement of the “Results” themselves. (1) Likely best left for the “Discussion”… But also (2) Not quite correct. The model results do not (conclusively) demonstrate that participants indeed “did” the task this way… But the model simulations presented are at least “qualitatively consistent” with what participants did and so the simulations do demonstrate at least one “plausible” way in which the experimental result may have arisen. But that does not preclude the alternative possibility that other models, based on other/different criteria / assumptions etc., might not also equally (or perhaps better) replicate the main features of the data. [Again, this seems more a topic for “Discussion” than “Results”…]

DISCUSSION:

LL. 498-522: The first 2 paragraphs offer no new information or insights – Just repetition. Could be removed, or at least greatly shortened / condensed.

LL. 524-537: See previous comment – Same here.

LL. 539-547: Similar to comments regarding the Introduction – The current narratives (Intro & Discussion) don’t (always) clearly articulate what is new, novel, different about *this* study relative to prior work…. Or why any such differences are “important”.

LL. 548-555: OK. So there was some difference between this study and Nayeem et al., 2021… Likely due to the differences in the protocols implements. Is this “important”? Or not? The current narrative does not address this.

L. 563: OK. Why is this “interesting”? Is it “important” in some way? This ball-cup paradigm is nonlinear – So nonlinearities in the viable solution spaces (defined as here or possibly in other ways) would not necessarily be surprising – No?

LL. 573-575: Again, is this finding particularly surprising? Humans perform many tasks under many contexts and set of constraints (physical constraints on the task itself, external constraints like time or metronome etc. – potentially many other things). Certainly, what objectives humans choose to prioritize can and do vary depending on many different aspects of a given task.

LL. 575-579: This idea (such as it is?) is just not very clearly developed. It is quite hard to discern if there is any real “conclusion” to be drawn here….

LL. 580-589: This section is rather weak. First, “This finding is consistent with several previous studies…” suggests that the present work really doesn’t add much “new” to what is already known in the context of this ball-cup manipulation task. Yes – Multiple prior papers from this same group working on (perhaps slightly different) variations of this task have presented quite similar conclusions.

LL. 580-589 (Cont.): Second, there is inherent circularity in the reasoning here. In the present work “relative phase stability” was defined and quantified in terms of relative phase “variability” (Fig. 5 & LL. 344-346, etc.). By definition though, things that are “less variable” are “more predictable” and so the sentence on LL. 584-585 is a circular statement. Hence, the concluding statement that “Humans might prefer predictable dynamics…” doesn’t add any new insight into the control of this task (or any other).

LL. 600-614: Again, not particularly clear. Not clear what this adds (that is new or different) to our understanding of these issues relative to several decades of prior work (that is described in this paragraph). The general conclusion (last sentence) is quite vague and highly speculative.

LL. 615-629: Again, interesting idea(s), but still somewhat vague and highly speculative [Yes – “future experiments” could be designed… But what exactly would that tell us? Why would that be “important”? Etc…]

LL. 630-658: See prior comments… Sure: “Further experiments and simulations… are required…” (etc.). But to what end?

LL. 659-673: Ending (again) with “Further studies are required…” – Same question(s) – See prior comments.

LL. 674-691: This time concluding with “Its an open question whether…” – But this still leaves the same question(s) – See prior comments.

Discussion – Overall: There is a lot of text here (8 pages total). But there is no coherent story that concretely describes what is new, what is important, or what the central contribution of the work conducted here is to our larger understanding of motor control. Clearly, a lot of work was done in conducting these experiments, analyses, and simulations.

METHODS:

Most all of this section largely repeats details already described in “Results” – Albeit with some additional details added here.

Two general comments – First, that the narrative throughout the Methods section would be improved by being very clear which aspects of these methods were repeated from prior studies (and giving proper citations etc.)… For example: “Following [Ref. ???], we…”, “Adopting the method of [Ref. ???], we…” etc. And to be very clear also where the approaches explicitly diverged from prior work… For example: “Unlike in [Ref. ???], here we…” etc.

Second, the section on “Modeling the Cup-ball System with Uncertainty in Pendulum Length” (LL. 847-861) essentially repeats (with a few additional details) that of the Results section (LL. 393-408). It still offers no clear reason why Gaussian noise was added to the pendulum length in the simulations, when no such noise was added to the (virtual) pendulum lengths in the experiments – although it very easily could have been (since the pendulum is virtual anyway). I suspect what the authors are trying to simulate here is not the experimental task itself, but rather the participant’s internal model of the task they were attempting to perform. If so, those are two very different things, and a critical distinction that is not made at all clear (either here in Methods, or before in Results).

**Have the authors made all data and (if applicable) computational code underlying the findings in their manuscript fully available?**

Reviewer #1: **No: **I do not see the Supporting Information the authors have submitted.

Reviewer #2: **No: **

Reviewer #3: **No: **They claim that "All relevant data are within the manuscript and its Supporting Information files".

That is just not adequate any more. No data are posted anywhere. No codes (neither analyses nor simulations) are posted anywhere. No indication that they intend to post any such data or code is given.

No.

PLOS authors have the option to publish the peer review history of their article (what does this mean?). If published, this will include your full peer review and any attached files.

Reviewer #1: No

Reviewer #2: No

Reviewer #3: No

**Figure resubmission:**
---

## [Decision Letter · Decision Letter 1]

7 Nov 2025

PCOMPBIOL-D-25-01193R1

Dynamical stability and mechanical impedance are optimized when manipulating uncertain dynamically complex objects

PLOS Computational Biology

Dear Dr. Lokesh,

Thank you for submitting your manuscript to PLOS Computational Biology. After careful consideration, we feel that it has merit but does not fully meet PLOS Computational Biology's publication criteria as it currently stands. Therefore, we invite you to submit a revised version of the manuscript that addresses the points raised during the review process.

We look forward to receiving your revised manuscript.

Kind regards,

Daniele Marinazzo

Section Editor

PLOS Computational Biology

**Additional Editor Comments:**

Dear authors

regarding the first comment of Reviewer 3, on the suitability of this paper for PLOS Computational Biology on the grounds of its contribution, my answer is "yes". 

Still, as Reviewer 3 correctly points out, the language needs to be revisited to properly present your results.

**Reviewers' comments:**

Reviewer's Responses to Questions

**Comments to the Authors:**

Reviewer #1: The authors have addressed all the points raised to my satisfaction. Excellent work!

Reviewer #2: The authors have carefully and successfully addressed all my previous comments. The revised manuscript demonstrates significant improvement in clarity, organization, and scientific rigor. The responses provided were detailed and appropriate, and the revisions made have adequately resolved the concerns raised in the earlier review. Overall, the manuscript is now suitable for publication in its current form.

Reviewer #3: PDF = “PCOMPBIOL-D-25-01193_R1_reviewer.pdf”

My Review:

The authors have largely done a thorough and thoughtful job of replying to the first round of reviewer comments. The manuscript has improved in many respects.

There are a few points though that I think still need to be addressed and that will further improve the final manuscript. I outline those below:

1. The “So What?” (i.e., Scientific Contribution) Question: I previously raised this issue that the manuscript does not make clear what the true contribution of this new work is and/or why or how it is an “important advance” (or equivalent term) of our knowledge of these topics. I think the authors largely missed my point. In their first response to the Editor Comments, they state: “1. We continue to argue that our research topic is understudied in computational motor control compared to other research topics. While the work presented here continues our investigation into complex object control, it unifies and extends prior work…” They expand on this response in subsequent responses. I fully agree that the topic of this paper is “understudied” and agree with the authors that there is a need for more work on manipulating complex objects. I think the line of work the PI (Sternad) has developed her over the last decade plus is a strong step in that process. I agree there are not many other papers from other research groups that try to tackle these types of problems, and I agree for the same reasons the authors present. But the question I raised was not whether there was enough work being done on our field on topics like this (I agree – not enough). The question was what new contribution does *this* paper make that goes beyond merely (incrementally) “extending prior work”? This new revised manuscript still does not really answer that question clearly.

That said, the question of whether these new results are “novel enough” compared to prior work is primarily an editorial decision, more so than a reviewer decision. So I will let the editors make the final call.

2. Formulation of Hypotheses vs. Predictions: The authors responded by stating that “… the distinction between hypothesis and prediction is far from being agreed upon in the community.” This is perhaps so, but in science in general, it has long been considered standard that “hypotheses” are only considered “scientific” if they are indeed "falsifiable" (https://en.wikipedia.org/wiki/Falsifiability). While the “hypotheses” presented here are insightful intuitive guesses (some indeed based on prior findings) as to the potential outcome of the experiment, the manuscript does not make clear if they are indeed falsifiable or not, nor what outcome/result would indicate if they were. Put another way, no viable alternative hypotheses are presented (e.g., in the form of something like “if theory A is correct, then result B is expected... Otherwise, if theory C is correct, then result D would be expected...” etc.). When the Abstract (L. 15; both previous and current) states that “The results supported… hypotheses:…” what follows is a summary of what the results *were*. The main manuscript is written in very much the same style. By any definition, a “hypothesis” is at least a *proposed* explanation (https://en.wikipedia.org/wiki/Hypothesis): i.e., it comes *before* the experiment is designed and *before* the results are known. What this manuscript presents is the exact opposite.

That said, I will reiterate what (I thought) I had tried to point out in my last review – which is to say that good science (empirical science in particular) need not always be “hypothesis driven”. I think this should be particularly true for journals like PLoS Computational Biology. The *computational* part of this work is why it was submitted to this journal in the first place – and it’s the modeling results here that help provide a plausible a posteriori explanation of the behavior that was *observed* in the experiment. It makes not one whiff of difference here whether that observed behavior was in tow to some “hypothesis” or not. If the model can help explain that behavior, so we can better understand it… Then that’s what the model is for.

Period.

In short – Don’t try to pound a square peg into a round hole. I think this manuscript would (could) be much stronger (IMO) if all of the (misleading) “hypothesis” language were removed and the narrative focused on the modeling and the ability of the model to explain the observed behavior… Period.

3. “Stability” – “Predictability” – “Variability” Still Conflated: I previously commented on the “inherent circularity in the reasoning…” of quantifying something called “relative phase stability” by calculating the standard deviation (i.e., “variability”) of relative phase and then tied these to the notion of “predictable dynamics”… The authors responded that “There is a subtle between stability and predictability…” and I completely agree!

The manuscript, however, continues to conflate these terms numerous times such that any distinction between them gets completely lost. Examples:

• L. 84: “By establishing stable and predictable…” – Leading to L. 90: “… to increase the predictability…”

• L. 91: “Stability of a nonlinear system…” – Leading to L. 98: “… ensuring simple and predictable…” – Leading to L. 101: “… achieve stable dynamics…”

• L. 137-138: “… important to stabilize the object…” – Then L. 141: “Human maximize stability…” [i.e., *not* “predictability”?] – Then L. 143: “… relative phase variability (stability)…”

• L. 708-709: “… stability was assessed as the variability of…”

• L. 732: “Stability Ensures Predictability…” – Leading to L. 736-737: “When the behavior… is stable, the dynamics… is more predictable…” – Leading to L. 740: “Humans might prefer predictable dynamics…”

Each of these statement tries to (in some way) tie one of these terms to the other. No definitions are given and none of the "subtle differences" the authors describe in their response letter are anywhere described in the manuscript. However, any of these terms can have different meanings and/or different definitions in different contexts. In some contexts, these can be *VERY* different. For example:

Center-of pressure (CoP) data during quiet standing are highly “unpredictable” (random), yet (when bounded to small amplitude, as typical for healthy adults) very “stable” (you don’t fall down or even become unbalanced).... OTOH: The simple inverted pendulum model is completely “predictable” (deterministic equations of motion), but highly “unstable” (falls over at the slightest perturbation).

Similarly for “variability” versus “predictability”: A simple since wave (e.g., SHO) with large amplitude is completely “predictable” but also highly “variable” (large SD) (and as the authors state in their response, "neutrally stable").... The "variability" (i.e., amplitude) of the sine wave is completely independent of its "stability". OTOH: Low-amplitude white noise is minimally “variable” (small SD), but completely un-predictable (random) (but could still be considered "stable" if it doesn't grow unbounded - like the CoP example). Etc.

So… It is the narrative of the manuscript itself that continues to go in circles between these three concepts without ever clearly defining any of them, or how they are either the same or different from each other. That specific definitions and distinctions are not presented in this manuscript will sow significant confusion among readers. The distinctions between these terms need to be clarified.

**Have the authors made all data and (if applicable) computational code underlying the findings in their manuscript fully available?**

Reviewer #1: None

Reviewer #2: None

Reviewer #3: Yes

PLOS authors have the option to publish the peer review history of their article (what does this mean?). If published, this will include your full peer review and any attached files.

Reviewer #1: No

Reviewer #2: No

Reviewer #3: No

**Figure resubmission:**
---

## [Editor Report · Decision Letter 2]

28 Nov 2025

Dear Dr. Lokesh,

We are pleased to inform you that your manuscript 'Dynamical stability and mechanical impedance are optimized when manipulating uncertain dynamically complex objects' has been provisionally accepted for publication in PLOS Computational Biology.

Best regards,

Daniele Marinazzo

Section Editor

PLOS Computational Biology

Daniele Marinazzo

Section Editor

PLOS Computational Biology

---

## [Editor Report · Acceptance letter]

PCOMPBIOL-D-25-01193R2

Dynamical stability and mechanical impedance are optimized when manipulating uncertain dynamically complex objects

Dear Dr Lokesh,

I am pleased to inform you that your manuscript has been formally accepted for publication in PLOS Computational Biology. Your manuscript is now with our production department and you will be notified of the publication date in due course.

With kind regards,

Anita Estes
